# Ice-stream flow switching by up-ice propagation of instabilities along glacial marginal troughs

Etienne Brouard[1]* & Patrick Lajeunesse[1]

[1] Centre d'études nordiques & Département de géographie, Université Laval, Québec, Québec

Correspondence to: Etienne Brouard (etienne.brouard.1@ulaval.ca)

**Abstract.** Ice stream networks constitute the arteries of ice sheets through which large volumes of glacial ice are rapidly delivered from the continent to the ocean. Modifications in ice stream networks have a major impact on ice sheets mass balance and global sea level. Reorganizations in the drainage network of ice streams have been reported in both modern and palaeo-ice sheets and usually result in ice streams switching their trajectory and/or shutting down. While some hypotheses for the reorganization of ice streams have been proposed, the mechanisms that control the switching of ice streams remain poorly understood and documented. Here, we interpret a flow switch in an ice stream system that occurred during the Pliocene-Pleistocene on the northeastern Baffin Island shelf (Arctic Canada) through glacial erosion of a marginal trough, i.e., deep parallel-to-coast bedrock moats located up-ice of cross-shelf troughs. Shelf geomorphology imaged by high-resolution swath bathymetry and seismostratigraphic data in the area points to the extension of ice streams from Scott and Hecla & Griper troughs towards the interior of the Laurentide Ice Sheet. Up-ice propagation of ice streams through a marginal trough is interpreted to have led to the piracy of the neighboring ice catchment that in turn induced an adjacent ice stream flow switch and shutdown. These results suggest that competition for ice discharge between ice streams, which implies piracy of ice-drainage basins via marginal troughs, was the driving mechanism behind ice flow-switching in the study area. In turn, the union of ice catchment by piracy increased the volume and discharge of Scott Ice Stream, allowing it to erode deeper and flow farther on the continental shelf. Similar trough systems observed on many other glaciated continental shelves may be the product of such a competition for ice discharge between catchments.

## 1 Introduction

Ice flow switching was first invoked to explain the recent stagnation of the Kamb Ice Stream—previously known as Ice Stream C—in West Antarctica Ice Sheet (Conway et al., 2002). Following this discovery, episodes of ice stream switching were also inferred to have occurred throughout the Pleistocene and in various palaeo-ice sheet settings (e.g., Dowdeswell et al., 2006; Vaughan et al., 2008; Winsborrow et al., 2012). Several driving mechanisms were proposed to explain the switching of ice streams: i) sediment accumulation causing a topographical change downstream (Dowdeswell et al., 2006); ii) local variations in bathymetry coupled with relative sea-level change (Stokes et al., 2009); iii) spatial and temporal variations in basal thermal

regime ( Ó Cofaigh et al., 2010); iv) presence of sticky spots, subglacial meltwater rerouting and ice thickness variations (Alley et al., 1994; Anandakrishnan et al., 2001; Anandakrishnan and Alley, 1997); v) competition for ice discharge and drainage basins (Greenwood and Clark, 2009; Payne and Dongelmans, 1997); and vi) topographic focusing (Sarkar et al., 2011; Storrar et al., 2017). The wide range of possible driving mechanisms outlines the fact that ice stream switching is a complex process that requires further assessment in order to model accurately the future behavior of modern ice sheets. Empirical data from the

palaeo-ice sheet record is therefore needed to better constrain the mechanism of flow-switching. Networks of glacial troughs on continental shelves may offer such insights for identifying and understanding the flow-switching mechanism as they were overdeepened by past ice streams. These networks generally consist of cross-shelf troughs often interconnected up-ice to marginal troughs (Anderson, 1999; Batchelor and Dowdeswell, 2014; Nielsen et al., 2005). While cross-shelf troughs are broadly aligned with fiords, marginal troughs are aligned parallel-to-coast. Marginal troughs are generally located along the

boundary between harder crystalline bedrock in the inner portion of the shelf and softer sedimentary rocks in the offshore portion of the shelf (Nielsen et al., 2005). They have been inferred to result from glacial erosion because they represent an up-ice extension of glacially eroded cross-shelf troughs (Nielsen et al., 2005), but no mechanism has been proposed to explain their formation. Here, high-resolution swath bathymetry imagery combined with archived seismic reflection data and International Bathymetric Chart of the Arctic Ocean  bathymetric data (IBCAO; Jakobsson et al., 2012) are used to analyze

the morphology and stratigraphy of a single glacial trough network on the northeastern Baffin Shelf, in Eastern Arctic Canada. These data suggest that past ice stream switching occurred due to the ice-discharge piracy from an ice stream in a deep cross-shelf trough (Scott Trough) via the lateral extension of a marginal trough (Hecla and Griper Trough), which led to the shutdown of the ice stream occupying the neighboring cross-shelf trough (Sam Ford Trough).

## 2 Regional setting and Glacial History

The analyzed troughs are located on the northeastern Baffin Island Shelf in Western Baffin Bay, Eastern Arctic Canada. The trough system is characterized by two cross-shelf troughs—Scott and Sam Ford—interconnected by one marginal trough: Hecla & Griper Trough (Fig. 1). Scott and Sam Ford troughs are aligned with the major fiords of northeastern Baffin Island: Scott Trough extends northeast from the mouth of Clark and Gibbs fiord while Sam Ford Trough extends northeast from Sam Ford and Eglinton fiords (Fig. 1). These four fiords were eroded in Precambrian crystalline bedrock that extends eastward

under the sedimentary bedrock strata. Precambrian basement is characterized under the shelf by half-grabens and grabens associated with rifting and spreading in Baffin Bay (Hosseinpour et al., 2013; Oakey and Chalmers, 2012). Post-rifting Cretaceous and Paleocene sedimentary rock strata overlay Precambrian bedrock on most of the shelf (Fig. 1) and are overlain by a relatively thin (<100 m) cover of unconsolidated Quaternary deposits (MacLean et al., 2014; Praeg et al., 2007). Previous studies have proposed that the cross-shelf troughs of Western Baffin Bay have a fluvial origin linked to a pre-Quaternary sea-

level lowstand (Fortier and Morley, 1956; Pelletier, 1964) much like the troughs of the Labrador Shelf (Josehans and Zevenhuizen, 1987). However, most of the erosion associated with the overdeepening—reaching up to 350–450 m in Scott

Trough—may be attributed to subsequent glacial erosion (Løken and Hodgson, 1971). The erosion of the troughs is intrinsically linked to the erosion of the deep fiords of northeastern Baffin Island, which modeling suggests were eroded to present-day depths in ~1 Myr (Kessler et al., 2008). The glacial overdeepening of the troughs probably began during the late Pliocene (~3.5 Ma; Srivastava et al., 1987) and all traces of preglacial fluvial systems were erased during the Pleistocene (Løken and Hodgson, 1971).

Not much is known on pre-Late Wisconsinan ice streams flowing through Baffin Island into Baffin Bay. To produce well-developed fiords, the position of these ice stream was probably stable throughout most of the Pleistocene, i.e., through few fiords and converging into fewer cross-shelf troughs. Glacial advances—and ice streams—of marine isotope stages (MIS) 5d/b and MIS4 were probably less extensive than during MIS2 (Ganopolski et al., 2010; Simon et al., 2014; Stokes et al., 2012); therefore, ice streams may not have reached the shelf edge between ~130 ka and 25 ka BP. The last glacial stage (MIS2) reached its maximum around 25 cal. ka BP in Western Baffin Bay, with the LIS reaching the shelf edge between Lancaster Sound and Home Bay (Fig. 1; Jenner et al., 2018). During the MIS2 (25 – 16 cal. ka BP), Scott and Hecla & Griper troughs were inundated by ice streams of the LIS (De Angelis and Kleman, 2007; Briner et al., 2006b; Brouard and Lajeunesse, 2017; Margold et al., 2015a) that extended to reach the shelf break at the mouth of the troughs, while Sam Ford Trough was under slow-flowing ice (Brouard and Lajeunesse, 2017).

Laurentide ice occupied most of the shelf until 14.1 cal. ka BP and deglaciation of the continental shelf was completed by ~15 – 12 cal. ka BP as coastal forelands emerged from the glacial ice cover (Briner et al., 2005, 2006a) and LIS outlets retreated to the fiord mouths after 14 ka cal BP (Brouard and Lajeunesse, 2017, 2019; Jenner et al., 2018). Paraglacial and postglacial sedimentation has been prevailing in the troughs from at least ~12 cal. ka BP and probably up to 14 cal. ka BP, which marks a minimum age for presence of outlets at the fiord mouths (Jenner et al., 2018; Osterman and Nelson, 1989; Praeg et al., 2007). Outlet glaciers of the LIS occupied the entire fiords until ~11.4 cal ka BP (Dyke, 2004) before rapidly retreating inland towards the fiord heads (Briner et al., 2009). Finally, glacial scouring observed inland of fiord heads indicates that (i) fast flowing ice streams may have extended inland of fiord heads and (ii) the fiords were efficient conduits for ice flow and erosion through the coastal mountain range of Baffin Island (Briner et al., 2008).

## 2 Methods

Swath bathymetric data collected using Kongsberg Simrad EM-300 (12 kHz) and EM-302 (30 kHz) multibeam echosounders onboard the CCGS Amundsen by the Ocean Mapping Group (University of New Brunswick) and the Laboratoire de Géosciences Marines (Université Laval) during 2003–2016 ArcticNet Expedition were used to analyze the seafloor of the Scott-Sam Ford trough system. The specifics of acquisition for each expedition can be obtained from Géoindex+ (geoindex-plus.bibl.ulaval.ca) and Ocean Mapping Group websites (omg.unb.ca/Projects/Arctic/ArcticMetadata.html). The multibeam bathymetry data were processed using Caris Hips & Sips and merged using the MB-System software, in order to provide a bathymetric surface at a 10 m-grid resolution for interpretation and analyses. The 10-m gridded surface was plotted over the

International Bathymetric Chart of the Arctic Ocean data (Jakobsson et al., 2012) to provide a complete, but lower resolution, coverage of the seafloor in between multibeam tracks. The complete surface was transferred in ESRI ArcMap 10.2 software geomorphological mapping and topographic analyses. Individual landforms were digitalized on the surface in ArcMap 10.2 and interpreted based on their apparent character (width, length, orientation, etc.) and on recent literature (e.g., Dowdeswell et al., 2016b). The interpretation of landform types is detailed in the results section.

We interpreted seismic reflection data from the Marine Data Holding public repository of National Resources Canada (Geogratis.gc.ca) in order to provide a subsurface view of the architecture of the shelf. Seismic reflection data were analyzed and extracted using the LizardTech GeoViewer software. Seismic reflection data were enhanced using the Brightness/Contrast tool in Adobe Photoshop CS5 for a clearer visualization. Maps and seismic reflection data were transferred to the Adobe Illustrator 2018 CC software for figure production and editing. BEDMAP2 data was used for analyzing Antarctica, in search of morphologically similar trough pattern systems and for map production (Fretwell et al., 2013).

## 3 Results

### 3.1 Troughs morphology

### 3.1.1. Scott Trough

Scott Trough is 62 km long, 12 km wide and 850 m deep. A longitudinal profile across the trough shows that it has a general trend of depths increasing from the shelf edge (~720 m) to the fiord sills (~900 m), forming a typical glacial overdeepening (Figs. 2–3). The trough is also characterized by a Precambrian bedrock sill that forms two bathymetric highs (~488 m) at about 15 km from the seaward extent of the trough (Figs. 1–2). The overdeepened basin between the Precambrian bedrock highs and the fiord sill is therefore overdeepened by at least ~360 m, when sediments are not considered. Sediment accumulations over bedrock in Scott Trough are thin, generally <10 m except for small patches in longitudinal basins showing up to 70 m of hemipelagic sediments, turbidites and mass failure deposits (Fig. 3). The sediments in Scott Trough basins were interpreted as relict (prior to the last glaciation, i.e., pre-Late Wisconsinan) deposits that have been preserved from glacial erosion (Praeg et al., 2007); alternatively, they could result also from ice-proximal sedimentation of an ice margin anchored at the fiord mouths, similar to deposits observed in Sam Ford Fiord (Brouard and Lajeunesse, 2019). The sediments in Scott Trough are mostly confined within longitudinal depressions that have up to ~130 m relief to bedrock (Fig. 3). The longitudinal depressions show erosional surfaces that indicate that they are not of structural origin. Airgun profiles on the ridges in-between the depressions show that although their upper portion consists of an unconsolidated unit, the basement is marked by strong reflections and hyperbola that are typical of bedrock. The overdeepening in Scott Trough is divided into 4 longitudinal basins separated by 3 bedrock and sediment ridges (Fig. 3). The 2 western basins are aligned with Clark and Gibbs fiords while the 2 eastern basins are prolonged into Hecla & Griper Trough. Scott Trough is bounded by steep sidewalls, longitudinal bathymetric highs (Fig. 4) and, at its seaward end, by a fan-shaped bathymetric bulge interpreted as a trough-mouth fan (Fig. 1; Brouard and

Lajeunesse, 2017). The transition from the fiords to Scott Trough is marked a steep wall and a drop of >450 m (Figs. 1–2). The transition from the fiords to the trough marks also the transition from the Precambrian crystalline basement to Sedimentary (Cretaceous and younger) bedrock (Fig. 1; Praeg et al., 2007).

### 3.1.2. Hecla & Griper Trough

Hecla & Griper Trough is 27 km long, 9 km wide and 780 deep. A longitudinal profile through the trough shows that it has a
general trend of depths decreasing from the junction with Scott Trough (~840 m) to Sam Ford Fiord Fiord sill (~450 m; Fig. 1). Hecla & Griper Trough is bounded on its southwest side by a steep wall of Precambrian crystalline bedrock (Figs. 1). The northeastern wall is less steep than the western wall and consists of sedimentary rock. The crystalline-sedimentary contact is not apparent on airgun data but, however, sedimentary structures and layers (parallel inclined reflectors) are apparent below sediments forming the seafloor of Hecla & Griper Trough, indicating that most of the rock underlying the trough is sedimentary
(Fig. 5). Hecla & Griper Trough can be divided into 2 longitudinal basins separated by a ridge that extends into Scott Trough. This latter ridge is composed of sediments without any bedrock ridge underneath, implying less geologic control over the development and position of the ridge, and is interpreted as a medial moraine (Fig. 5). Other similar ridges occur in Hecla & Griper Trough but are carved in sedimentary bedrock. The transition between Hecla & Griper Trough and Sam Ford Fiord is marked by a 12 km large and 350–400 m deep plateau to the southwest, by bedrock shoals to the northeast and a 1.5 km-wide
depression in-between (Fig. 1).

### 3.1.3. Sam Ford Trough

Sam Ford Trough is 77 km long, 13 km wide, and 370 m deep. Similarly to Scott Trough, the bathymetric profile through Sam Ford Trough shows generally increasing depths landward, but only from the middle shelf. However, the middle shelf bathymetric high in Sam Ford Troughs is not of bedrock origin, but coincides with an accumulation of glacial sediments
constructed during the stabilization of an ice margin (Fig. 6; Praeg et al., 2007). The bedrock excavation of Sam Ford Trough exceeds by at least 75 m modern water depths for most of its length and is overdeepened by about 200 m landward. The accumulation of glacial sediments appears on the swath bathymetry imagery as 3 distinct lobes with one broadly aligned parallel to the trough. Sam Ford Trough is bounded laterally by longitudinal ridges that extend from Sam Ford Fiord sill up to the shelf break (Figs. 1–7). The sill at the mouth of Sam Ford Fiord marks the transition where the fiord becomes the trough.
This sill is characterized by a >470 m depression to the south (Fig. 1) and plateau to the north. The southern depression extends to form Sam Ford Trough. The plateau on the northern part of the sill is mainly characterized by crystalline bedrock ridges and flat sediment-filled basins.

### 3.2 Ice-flow landforms

#### 3.2.1. Crag-and-tails

Crag-and-tails are flow-oriented positive landforms with an identifiable bedrock "crag" at the head and a drift tail (Evans and Hansom, 1996). A total of 202 crag-and-tails occur within the trough system and on Sam Ford Fiord sill (Fig. 7). The crag-and-tails have lengths ranging between 98 m and 5935 m with a mean length of ~1071 m. They occur between 254 m and 836 m depths and most of them are located seaward of fiord sills and seaward of bedrock highs in Scott Trough (Bennett et al., 2014). Crag-and-tails have been widely used in paleoglaciological reconstructions as indicators of ice-flow orientation (e.g., (Brouard et al., 2016; Brouard and Lajeunesse, 2017; Hogan et al., 2010; Jansson et al., 2003; Kleman et al., 2007).In association with mega-scale glacial lineations, drumlins and grooves, the presence of crag-and-tails generally indicates fast ice-flow conditions, i.e., ice streaming. Overdeepened curvilinear depressions (crescentic scours) occur in some cases upstream of crag-and-tails (Fig. 8). The presence of crescentic scours in front of crag-and-tails probably indicates the presence of meltwater (Graham et al., 2009; Graham and Hogan, 2016).

#### 3.2.2. Drumlins

Drumlins are smooth, asymmetric, oval-shaped hills (positive landforms) with a steeper stoss side and a more gentle-sloping lee side (Clark et al., 2009). These landforms occur at depths ranging from 179 m to 755 m. The 486 mapped drumlins have length ranging between 78 m and 1856 m with a mean of ~ 357 m, and are mostly located on the sills of the fiords (Fig. 7). Drumlins have a long axis oriented parallel to ice flow and occur in clusters and in association with glacial lineations, crag-and-tails, whalebacks and meltwater channels. Grouped into flow sets, drumlins can reveal palaeo-ice-flow orientation. Patterns of elongation and convergence of drumlins (together with other ice-flow landforms such as mega-scale glacial lineations and crag-and-tails) can be used as a relative indicator of ice-flow velocity and can therefore be used to identify ice-stream tracks (Stokes and Clark, 2002a). Drumlins are formed under wet-bed glaciers (King et al., 2007) and occur in association with meltwater-related landforms such as meltwater channels or crescentic scours (Fig. 8). Overdeepened curvilinear depressions (crescentic scours) are in some cases present upstream of drumlins (Fig. 8).

#### 3.2.3. Glacial lineations or mega-scale glacial lineations (MSGL)

Glacial lineations are highly elongated (apparent elongation ratio 1: 10) parallel ridges (positive landform) formed in glacigenic sediments (Clark, 1993). Most glacial lineations are observed in Scott and Hecla & Griper Trough (Fig. 7) where they have lengths ranging from 156 m to 8664 m with a mean of ~1026. Glacial lLineations occur in the troughs at depths between 218 m and 840 m. They are generally oriented parallel to the troughs. Usually observed in sets, these landforms have a soft and mostly regular texture that likely reflects a sedimentary character (Ó Cofaigh et al., 2005, 2013). In some cases, ridges have a rougher texture that may imply a bedrock character or very thin sediment cover. MSGLs are indicators of fast ice flow suggesting palaeo-ice stream activity (Clark, 1993; Stokes and Clark, 2002a). Accordingly, they also indicate ice-flow

orientation. Cross-cut by grounding-zone wedges, mega-scale glacial lineations are interpreted to reflect time-transgressive ice flows occurring during the landward retreat of an ice stream (Brouard and Lajeunesse, 2017; Dowdeswell et al., 2008).

### 3.2.4. Grooves

Grooves are linear to curvilinear negative landforms observed both in sediments and in bedrock. Grooves usually occur in association and aligned with mega-scale glacial lineations, crag-and-tails and drumlins. Alike the lineations, the grooves were mostly mapped in Scott and Hecla & Griper troughs (Fig. 7). The 465 mapped grooves have lengths ranging between 84 m and 5882 m, with a mean of 944 m. They occur at depths between 127 m and 838 m. Produced by keels beneath glacial ice eroding the underlying substrate, grooves record palaeo-ice flow direction (Graham et al., 2009; Livingstone et al., 2012). Their presence alongside MSGLs under present-day ice streams suggests that they are the product of fast ice flow, i.e., an ice stream (Jezek et al., 2011; King et al., 2009).

### 3.2.5. Ice stream lateral moraines (ISLMs)

Ice stream lateral moraines are curvilinear ridges (positive landform) observed on the sides of cross-shelf troughs. Ice stream lateral moraines are characterized by a gentle slope on their trough side and a steeper shelf side (Batchelor and Dowdeswell, 2016). Ice-stream lateral moraines are present on both sides of cross shelf troughs, but are not observed on the sides of Hecla & Griper Trough, probably because of the lack of decent data (Figs. 1–7). ISLMs in the study area are up to 61 km-long and 8 km wide. Ice-stream lateral moraines are believed to be formed subglacially at the shear zone between fast-flowing ice and slower-flowing ice or ice-free terrain (Batchelor and Dowdeswell, 2016). They can therefore be used to delineate the lateral extent of an ice stream (Brouard and Lajeunesse, 2017; Margold et al., 2015b; Stokes and Clark, 2002b).

### 3.2.6. Meltwater channels

On swath bathymetry imagery, meltwater channels take the shape of sinuous longitudinal depressions (negative landform) that are generally carved in bedrock (Lowe and Anderson, 2003; Nitsche et al., 2013; Slabon et al., 2018). The 235 mapped meltwater channels occur at depths ranging between 202 m and 718 m (Fig. 7). Their length varies from 126 m to 4.1 km. Some channels are characterized by a flat bottom that indicates sediment infill (Brouard and Lajeunesse, 2019; Smith et al., 2009). They form anastomosing networks often extending in-between ice-flow landforms (MSGLs, drumlins, crag-and-tails). The presence of these channels indicates abundant meltwater that could favor ice-bed decoupling, enable basal sliding, and generate ice streaming (Anandakrishnan and Alley, 1997; Engelhardt et al., 1990; Lowe and Anderson, 2003; Reinardy et al., 2011).

### 3.2.7. Subglacial medial moraines

Subglacial medial moraines are large curvilinear sediment ridges (positive landform) with a smooth character and are ice flow oriented. On swath bathymetry imagery, they can be overprinted by iceberg ploughmarks, MSGLs and grooves. Subglacial

medial moraines are thought to be formed subglacially under constraints created by coalescing glaciers (Dowdeswell et al.,
2016a). They therefore reflect the downstream movement of ice and act as indicators of ice-flow direction. Subglacial medial
moraines were first identified in Scott Trough (Fig. 7; Dowdeswell et al., 2016a). Subglacial medial moraines were mapped in
Scott and Hecla & Griper troughs (Fig. 7), where they reach up to 55.5 km in lengths.

### 3.2.7. Whalebacks

Whalebacks are rough, asymmetric and oval-shaped hills (positive landforms) with a steeper stoss side and a more gentle-
sloping lee side (Evans and Hansom, 1996; Roberts and Long, 2005). A total 321 whalebacks were identified. They occur at
depths between 170 m and 700 m and have lengths varying between 61 m and 1044 m. The whalebacks have a flow-oriented
long axis and are generally observed in clusters. In the study area, they occur in association with glacial lineations, crag-and-
tails, drumlins and meltwater (Fig. 7). The whalebacks, grouped into flow-sets can be used to reveal palaeo-ice-flow direction
(Krabbendam et al., 2016). Eroded into bedrock they can result of multiple glacial erosion cycles and therefore record multiple
ice flows. Alike for drumlins and crag-and-tails, overdeepened curvilinear depressions (crescentic scours) can also be present
upstream of some whalebacks.

### 3.3 Grounding-zone wedges

Grounding-zone wedges (GZW) are asymmetric tabular bathymetric wedges (positive landform) that are perpendicularly
aligned to trough or fiord orientation. GZW are characterized on swath bathymetry imagery by an extensive stoss side with
low gradients and a steeper and narrower lee side. Grounding-zone wedges occur in Scott (n=4) and Sam Ford troughs (n=3),
forming bathymetric mounts (Figs. 1–7). GZW are formed by the accumulation of subglacial sediments at the grounding zone
of an ice stream during temporary standstills of an ice margin (Dowdeswell and Fugelli, 2012; Lajeunesse et al., 2018). They
have also been associated with the presence of ice shelves (Dowdeswell and Fugelli, 2012). The presence of an ice-shelf is
believed to restrict vertical accommodation space for sediments in favor of sediment progradation, which explains the low-
amplitude and horizontally extensive (up to 139 km$^2$) character of GZWs.

## 4 Discussion

### 4.1 Ice stream tracks

In Scott Trough, the presence of mega-scale glacial lineations (MSGLs) extending to the shelf break, together with a till unit
extending on the trough-mouth fan and glacial debris flows on the shelf edge, indicate that the Laurentide Ice Sheet (LIS)
reached the shelf break during the Last Glacial Maximum (LGM; Brouard and Lajeunesse, 2017; Jenner et al., 2018). MSGLs,
crag-and-tails and drumlins mapped in this region indicate that during the LGM, a tributary ice stream of the Scott Ice Stream
extended from Sam Ford Fiord to Scott Trough through Hecla & Griper Trough (Brouard and Lajeunesse, 2017). Accordingly,
the medial moraine ridges are interpreted to be the product of differential ice-stream erosion. In a same way, coalescence of

multiple ice streams probably favored the formation of subglacial medial moraines over the ridges (Dowdeswell et al., 2016a).

While the glacial origin of Scott Trough is obvious from its overdeepening and numerous ice-flow landforms, the glacial origin of Sam Ford Trough is confirmed by the presence of (1) ice stream lateral moraines and (2) grounding-zone wedges in the trough and a small overdeepened basin on Sam Ford Fiord sill (Figs. 1-6-7-8). However, the absence of ice-flow landforms in Sam Ford Trough suggests that the trough was occupied by slow-flow ice during the last glacial episode. The slow ice flow contrasts with a recent glacial reconstruction stating that in full glacial conditions, Sam Ford and Scott troughs were characterized by separate ice streams that drained approximately comparable areas from the LIS (Batchelor and Dowdeswell, 2014). The presence of ISLMs on the sides of Sam Ford Trough together with an overdeepened basin and a 75 m-thick GZW (Figs. 6-7) in the trough also indicate that an ice stream originating from Sam Ford Fiord has also been, at some point, efficient enough to excavate the trough and to flush enough subglacial sediment to construct the multiple GZWs. However, the ice streams in Sam Ford Trough were not efficient enough to build a trough-mouth fan at the seaward end of the trough. The general thickness of paraglacial and postglacial sediments (~40 m) in Sam Ford Trough also suggests that the GZWs are older than the last glaciation (i.e., pre-Late Wisconsinan) and confirms the occurrence of slow-flowing and less erosive ice in Sam Ford Trough during the last glaciation. In comparison, paraglacial-postglacial sediments in Scott Trough, outside of the basins, never exceed 10 m in thickness.

Glacial bedforms on the sill of Sam Ford Fiord indicate that an ice stream flowed from Sam Ford Fiord through Hecla & Griper Trough. Although Hecla & Griper Trough has a structural origin since it was eroded along the crystalline-sedimentary contact, glacial lineations, crag-and-tails and grooves within the trough and on the sill of Sam Ford Fiord indicate that it was eroded by an ice stream (Brouard and Lajeunesse, 2017). The bathymetric profile in Hecla & Griper Trough (Fig. 2) showing depths increasing towards Scott Trough implies that ice-stream erosion was the most effective at the junction between Hecla & Griper and Scott troughs.

## 4.2 Erosion of Hecla & Griper Trough

Erosion of fiords, troughs and bedrock basin overdeepenings has been proved to be a function of topography, ice discharge, subglacial hydrology, basal thermal regime, basal ice debris and glacier size (Cook and Swift, 2012; Kessler et al., 2008; Ugelvig et al., 2018). Overall, higher velocities together with thicker ice will generate more basal melt and will accelerate basal sliding and sediment flushing (e.g., Cook and Swift, 2012; Jamieson et al., 2008). Kessler et al. (2008) also demonstrated that topographic steering along was sufficient to produce overdeepened fiords and troughs. Analyses of bedrock basins of Antarctica and Greenland also indicated a similar erosion process where ice-flow confinement leads to deeper overdeepenings (Patton et al., 2016). This suggests that if an ice stream—and the associated erosive power—originated from Sam Ford Fiord, it should be expected that at equivalent ice velocity Hecla & Griper Trough would be the deepest at its narrowest section, i.e., in-between the shoals and the shelf. However, Hecla & Griper Trough is the deepest at its widest, i.e., at its junction with Scott Trough. The erosion of Hecla & Griper Trough was therefore more dependent on down-ice dynamics occurring in Scott Trough rather than up-ice in Sam Ford Fiord. The down-ice dependence of Hecla & Griper Trough erosion is in agreement with an

inferred strong relationship in the development of interconnected marginal troughs and cross-shelf troughs (Nielsen et al., 2005). Coincident with the contact between crystalline and sedimentary bedrock, Hecla & Griper Trough is therefore interpreted to result from a tributary ice stream of Scott Ice Stream eroding sedimentary bedrock along a weakness line, resulting in a morphology that is similar to other marginal troughs (Anderson, 1999; Nielsen et al., 2005). The down-ice dependence of erosion in Hecla & Griper also indicates that most effective erosion occurred in an up-ice direction. Similarly, analyses of overdeepenings in Antarctica and Greenland showed that below sea-level, most of the erosion of overdeepening occurs by headward erosion (Patton et al., 2016). Upstream propagation of high velocities—and erosional power—in ice streams is mainly dependent of instabilities at the ice margin or at the grounding zone; i.e., an acceleration of ice flow can propagate up-ice the main trunk of the ice stream and up-ice in the tributary ice streams (De Angelis and Skvarca, 2003; Hughes, 1992; Kleman and Applegate, 2014; Payne et al., 2004; Retzlaff and Bentley, 1993). Such up-ice progression of an ice stream can create a positive ice-erosion feedback, in which erosion of the overdeepening causes the headwall to steepen and therefore further enhances sliding velocity and erosional power up-ice the ice stream (Herman et al., 2011). Therefore, Hecla & Griper Trough as a host to a tributary ice stream can have facilitated upstream propagation of ice streaming up to Sam Ford Fiord sill, forming a mature marginal trough.

### 4.3 Ice stream switching

The presence of ice-flow landforms on Sam Ford Fiord Sill that are oriented towards Hecla & Griper Trough implies that ice discharge through Sam Ford Fiord was not oriented towards Sam Ford Trough. Sam Ford Trough is, however, interpreted as the product of an ice stream mainly originating from Sam Ford Fiord and in a minor way from Eglington Fiord (Margold et al., 2015a). Eglington fiord is interpreted as a minor contributor to Sam Ford Ice Stream because of the presence of a GZW (medial subglacial moraine?) attached to the junction between the fiords and clearly indicating that most of the ice discharge came from Sam Ford Fiord (Fig. 7). A possible scenario for the occurrence of landforms associated with an ice stream flowing from Sam Ford Fiord to both Scott and Sam Ford troughs would be that ice flow from Sam Ford Fiord was partitioned between Scott and Sam Ford troughs. Recent research suggests that if Sam Ford and Scott troughs were both occupied by comparable ice streams during full glacial conditions, they would have had similar drainage areas (Batchelor and Dowdeswell, 2014), similar width and therefore similar ice discharge (Stokes et al., 2016). The medial moraines in Scott Trough indicate that half of the ice that eroded the trough came from Sam Ford Fiord. Therefore, ice from Sam Ford Fiord would be responsible for about half of Scott Trough excavation, while also being responsible for the excavation of Sam Ford Trough. Accordingly, ice discharge through Sam Ford Fiord would need to have been twice the ice discharge of combined neighbouring Clark and Gibbs fiords. However, a 2:1 ratio of Sam Ford ice discharge over and Scott ice discharge is not compatible with estimates of paleo-ice discharge (1.04:1) and drainage area (1:1) that can be derived from recent models (Batchelor and Dowdeswell, 2014; Stokes et al., 2016). Another possible scenario that could explain the occurrence of landforms related to an ice stream flowing from Sam Ford Fiord to both troughs would be that Sam Ford Fiord ice discharge has always been directed towards Hecla & Griper Trough and that Sam Ford Fiord is the product of most extensive glaciations. During most extensive glaciations, a part of Sam

Ford Fiord ice discharge could be spilled towards Sam Ford Trough to form a shallower trough. This is unlikely to be the case because the last glaciation (MIS-2) was the most extensive glaciation in North America since MIS-6 (<130 ka BP; Ganopolski et al., 2010; Simon et al., 2014; Stokes et al., 2012) and one of the most extensive throughout the Pleistocene (Ehlers and Gibbard, 2003); yet there is no evidence for an ice stream in Sam Ford Trough during the last glacial episode. Also, if ice discharge from Sam Ford Fiord has always been directed towards Hecla & Griper Trough, it is expected that ice would erode,

by topographic steering, an overdeepening at the narrowest section of Hecla & Griper Trough (See section 4.2). However, it is possible that a small part of Sam Ford ice discharge could have been advected towards Hecla & Griper Trough from its very beginning and facilitated ice piracy by promoting the up-ice propagation of high velocities into an already fast-flowing ice system.

Therefore, partitioned ice discharge from Sam Ford Fiord in both troughs is possible but it cannot account for the
320 trough morphology and for the full-scale erosion of both Sam Ford and Scott troughs. It is easier to propose a mechanism where ice from Sam Ford Fiord switched from one trough to the other. Sam Ford Ice Stream would have first eroded Sam Ford Trough and then switched orientation to flow through Hecla & Griper Trough. Following this ice flow-switching, the erosion in Sam Ford Trough ceased while the erosion in Scott Trough was enhanced by increased ice discharge; the difference in erosion volume between San Ford and Scott troughs can thus be explained by this mechanism.

As mentioned earlier, several mechanisms have been proposed to explain the switching behavior of an ice stream. Here we propose a mechanism that incorporates some of these mechanisms but where the flow switch of Sam Ford ice stream is due long-term erosion of up-ice propagating ice streams (i.e., Scott Ice Stream). Ice streams propagating upstream and eroding Hecla & Griper Trough have eventually eroded the marginal trough up to a point where it extended up-ice to reach the Sam Ford Fiord mouth. The changes in fiord depth-to-bedrock resulting from the erosion of the marginal trough and associated
with the upstream propagation of Scott Ice Stream in Sam Ford system led to the: 1) reorganization of the ice drainage system; 2) switching of Sam Ford Ice Stream from Sam Ford Trough to Scott Trough; and ultimately, 3) shutdown Sam Ford Ice Stream in Sam Ford Trough(Alley et al., 1994; Anandakrishnan and Alley, 1997; Graham et al., 2010)(Alley et al., 1994; Anandakrishnan and Alley, 1997; Graham et al., 2010). Such ice piracy through the switching of ice streaming most probably occurred early during Pliocene-Pleistocene glaciations; it could explain the absence of a trough-mouth fan at the seaward end
of Sam Ford Trough and also why Sam Ford Trough is more than two-times shallower than Scott Trough. It is probable that ice streaming in Sam Ford Trough only occurred during a short period at the beginning of early (Late-Pliocene—Early Quaternary?) glaciations, before its ice discharge was captured by the Scott Ice Stream. As this process repeated itself throughout glacial cycles, it accentuated the depth of Hecla & Griper Trough, which in turn facilitated the capture of Sam Ford Ice Stream by Scott Ice Stream during subsequent glaciations. The changes in depth-to-bedrock associated with the upstream
propagation of Scott Ice Steam in Hecla & Griper also led to a point in time when the Sam Ford Fiord ice discharge switched to be topographically diverted towards Hecla & Griper (i.e., the 1.5 km-wide depression). From that point on, ice-streaming conditions were unlikely to occur in Sam Ford Trough. Taken together, these observations suggest that the erosion and the morphology of the troughs of northeastern Baffin Shelf is a function of a competition for ice drainage basins. This process

shares striking similarities with fluvial drainage-basin reorganizations linked to river captures (Bishop, 1995): where in the northeastern Baffin Island case, a dry valley (Sam Ford Trough) with fluvial deposits (here glacial, i.e., the GZW) and a knickpoint (the 1.5 km-wide depression at the head of Hecla & Griper Trough).

An ice-drainage piracy mechanism similar to river captures in fluvial systems can probably not explain the occurrence of all the other abandoned cross-shelf troughs that do not extend up-ice to the coast or to fiords, mainly because each have different geological, glaciological and climatic contexts. However, the presence of morphologically similar systems can be observed on many formerly glaciated continental shelves: e.g., small troughs along the Antarctic Peninsula (Fig. 9a) and Pennell Trough on West Antarctica Shelf; Unnamed Trough on Disko Bank on West Greenland Shelf (Fig. 9b); Okoa Bay on the Baffin Shelf (Fig. 9c); small cross-shelf troughs off the coast of Labrador and Newfoundland (Canada; Fig. 9d); or Angmagssalik Trough on East Greenland Shelf. Much alike Sam Ford Trough, these cross-shelf troughs are shallow and oriented towards a fiord that probably fed an ice stream during full glaciation. However, these shallower troughs do not reach the fiord mouth because they are all intersected by the marginal trough of a deeper adjacent cross-shelf trough reaching the shelf break, suggesting that (i) they were cut-off from their glacial ice supply and (ii) their ice stream was probably shutdown, as with the Sam Ford Ice Stream. The presence of the such abandoned and less eroded cross-shelf troughs—which are also intersected by marginal troughs—on most formerly glaciated continental shelves of the world possibly result from a similar mechanism to Scott-Sam Ford system. However, further investigations are needed to confirm that competition between ice stream played a role in the development of these trough systems.

## 4.4 Competition for drainage basins and ice sheet stability in marine environments

Although competition for ice-drainage basins between ice streams has long been recognized as a driver for changes in ice sheet geometry (Greenwood and Clark, 2009; Payne and Dongelmans, 1997), it has not been given much attention compared to other hypotheses for switching as this idea has only been derived from modeling studies. However, the geomorphology, pattern and glacial trough assemblages in the Scott-Sam Ford system provide for the first time an empirical context where adjacent ice streams on a continental shelf could have interplayed in a competition for ice discharge during the Pliocene-Pleistocene. The erosion profile of Hecla & Griper Trough indicates that (i) this competition was dependent of glaciodynamics occurring down-ice, probably at the marine-based ice margin, and (ii) the upstream propagation of ice instabilities (or ice acceleration) occurring down-ice play a fundamental role in organizing ice-flow routes within the ice sheet. Felikson et al. (2017) modeled such up-ice propagation of a wave of thinning linked to changes in ice margin thickness. The hypothesis of instability waves propagating upstream in ice streams and outlet glaciers has also been mentioned in a series of recent papers (e.g., Felikson et al., 2017; Nick et al., 2009; Price et al., 2011; Reese et al., 2018). The erosion of the Hecla & Griper marginal trough confirms such a wave theory: instability waves traveling up-ice were redirected along the line of weakness along sedimentary-crystalline contact, which enabled the formation of a tributary ice stream that eroded Hecla & Griper Trough. Redirection of the instability wave was probably due to: i) differential erosion and ii) the competition between advection and diffusion of the wave (Felikson et al., 2017). Differences in bedrock lithologies most likely favored differential erosion rates that eventually resulted in a

topographic step (steep bed profile) at the transition between crystalline and sedimentary bedrock. This topographic step favored relatively thin ice and a steep ice profile, where down-ice advection dominated and thus limited up-ice diffusion of ice streaming from that location. Conversely, the cross-shelf trough forms a relatively flat bed with flat-ice profiles where diffusion

of the wave could have dominated. In this setting, the diffusion of the wave was redirected on the shelf along the transition between crystalline and sedimentary bedrock where the bed profile was the flattest. Long-term erosion along the bedrock contact could then have led to the erosion of a marginal trough. The development of a marginal trough was a key structural element to produce a positive feedback that facilitated redirection and diffusion of up-ice acceleration. The competition between adjacent ice streams on the continental shelf was therefore "won" by the ice stream that was the most efficient at

diffusing up-ice its fast flow through its tributaries, i.e., through a marginal trough.

The competition between ice streams and their switching on and off imply important configuration changes in ice stream and ice sheet geometries. As inland thinning of ice sheets can be controlled by ice margin geometry (Felikson et al., 2017), the competition for the ice drainage basin represents a major driver in the growth and decay of an ice sheet. Accordingly, abrupt changes in ice stream networks are also expected to influence ice sheet stability. Where ice stream switching occurs

and lead to the merging of two ice streams, ice-flow acceleration can propagate upstream in both glacial drainage basins (i.e., in both Scott and Sam Ford in the study area). Merged ice streams can then affect a greater area of the ice sheet and provoke ice-divide migration (Greenwood and Clark, 2009). Such flow-switching of ice streams can provide more efficient and rapid pathways for continental ice to reach the ocean, possibly leading to a more rapid drawdown of the ice sheet. The merging of ice streams through ice piracy should result in an increase in ice discharge and erosion rates in the "winning" trough. The

winning ice stream gains mass balance and thus should equilibrate by advancing its margins (if it is not already at the shelf break).

Although the overdeepened bedrock basin of Scott Trough and its seaward extent to the shelf break cannot be unequivocally attributed to ice stream-switching, the longitudinal basins in Scott Trough that are related to ice flow from Sam Ford Fiord represents almost half of the width of Scott Trough. The presence of these basins suggests the switching could

account for up to half of the erosion of Scott Trough. The mechanism of ice stream switching through marginal troughs can therefore lead to more extensive glaciations and to the erosion of deeper glacial troughs on high latitude continental shelves: merging ice streams would allow the glacial ice to extend farther and erode deeper on these shelves. The increasing depths in troughs behind grounding zones are known to create a positive feedback during glacial retreat that result in more rapid retreat of the ice margin in progressively deeper waters (Joughin and Alley, 2011; Mercer, 1978; Schoof, 2007). Therefore, a flow

switch caused by marginal trough erosion is expected to make ice sheets occupying similar trough systems on continental shelves more sensitive to climate forcing, on both short (within a glacial cycle) and long term (over multiple glacial cycles).

## 5 Conclusions

The swath bathymetry imagery and the geophysical data collected in a simple trough network of the Northeastern Baffin Island Shelf (Arctic Canada) provide evidence for the flow switch of a ice stream that occurred during the Pliocene-Pleistocene on the northeastern Baffin Island shelf (Arctic Canada) through glacial erosion and overdeepening of a marginal trough. The glacial geomorphology of the seabed and the erosion profiles of the troughs network provides for the first time an empirical context where adjacent ice streams could have interplayed in a competition for ice discharge. This competition for ice discharge between ice streams was dependent of dynamics occurring at the marine-based ice margin (i.e., the grounding zone), indicating that the upstream propagation of high glacial velocities plays a fundamental role in organizing ice-flow routes and catchments within ice sheets. Such a union of glacial ice catchments by switching provides more efficient and rapid pathways for continental glacial ice to reach the ocean, enhancing the extent and erosive action of ice streams on high latitude continental shelves. These results suggest that the up-ice propagation of high ice velocities is a major driver of changes in the geometry of ice sheets during their growth and decay; it can therefore affect ice sheet stability. The hypothesis that the ice stream switching mechanism plays a major role on the geological evolution of formerly glaciated continental shelves through the extension of marginal troughs is supported by other examples from continental shelves located at the former margins of the Laurentide, the Greenland and the West Antarctica ice sheets. These trough systems show striking morphological similarities with the through networks described in the study area and could also result from a competition between adjacent ice streams. As Laurentide Ice Sheet palaeo-dynamics inferred from Canadian Arctic continental shelves can be used as an analogue to understand how modern marine-based ice-sheets will respond to future climate change and sea-level fluctuations, these results highlight the need for further investigations and further modeling studies that should include ice-drainage competition between ice streams and the resulting flow capture as a major control on ice sheet stability.

## 6 Acknowledgements

We sincerely thank the captains, crew, and scientific participants (particularly Gabriel Joyal, Annie-Pier Trottier and Pierre-Olivier Couette, Université Laval) of ArcticNet cruises 2014–2016 on board the CCGS Amundsen. We also thank John Hughes Clarke and his team at the Ocean Mapping Group (University of New Brunswick) who collected swath bathymetry data in the region between 2003 and 2013. We thank 2 anonymous reviewers for providing valuable comments that improved the quality of the manuscript. This project was funded by ArcticNet Network Centres of Excellence and NSERC Discovery grants to P.L.

## 7 Authors contribution

E.B. and P.L. developed the study. E.B. interpreted the geophysical datasets, wrote the paper, and prepared the figures. P.L. helped with the interpretation and analysis, and contributed to the writing and editing of the paper.

## 8 Competing Interests

The authors declare that they have no competing interests.

## 9 Data availability

The multibeam bathymetry dataset can be visualized and requested on the Université Laval Géoindex+ website (geoindex-plus.bibl.ulaval.ca). The seismic reflection data along with the acquisition specifics are available on the Geological Survey of Canada website (ftp.maps.canada.ca/pub/nrcan_rncan/raster/marine_geoscience/Seismic_Reflection_Scanned/).

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

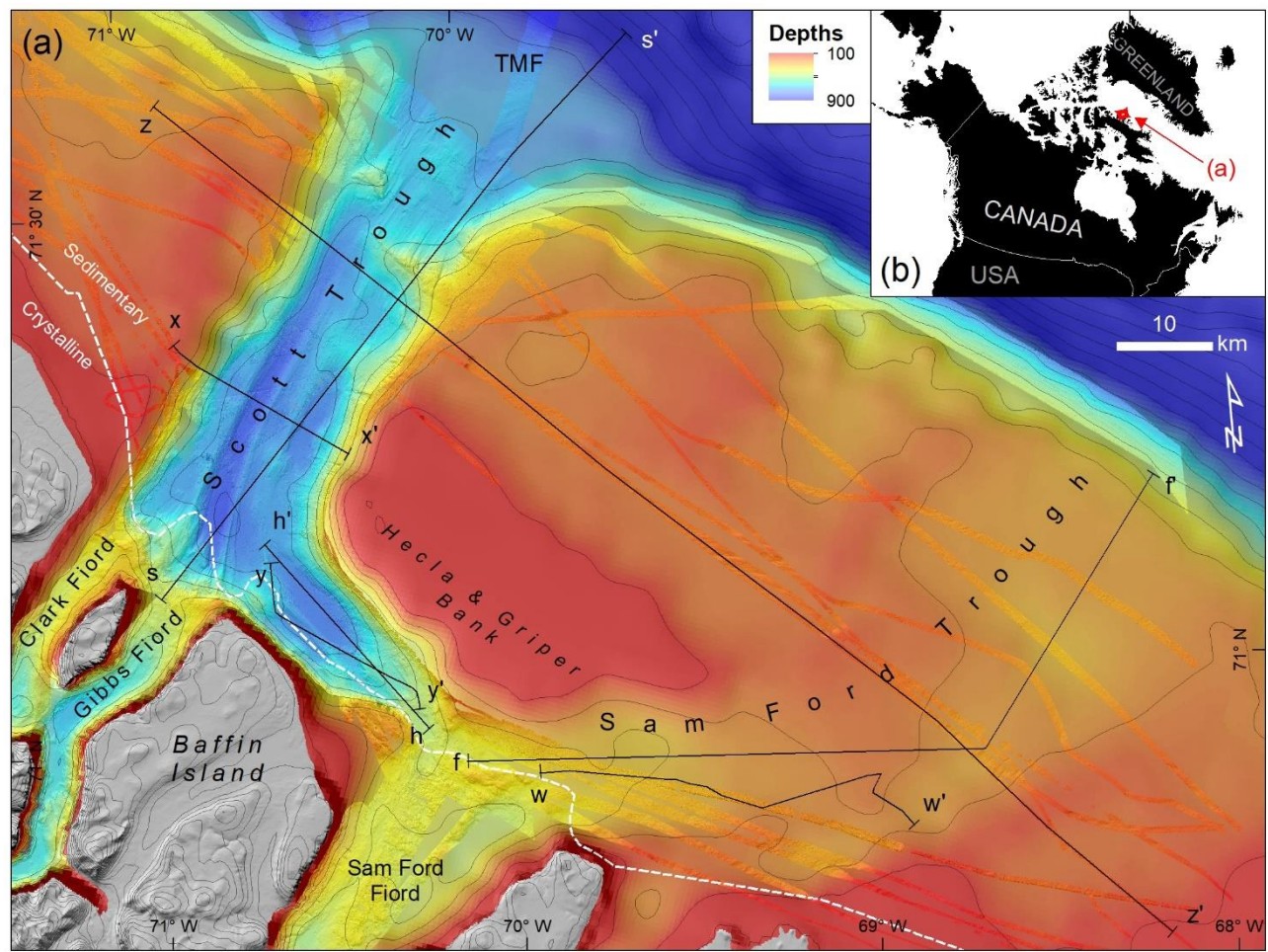

**Figure 1 A. Map showing the high-resolution bathymetric data collected by ArcticNet program (2003–2016) draped on the International Bathymetric Chart of the Arctic Ocean data (IBCAO; Jakobsson et al., 2012) map on the northeastern Baffin Island**
**shelf. The black dashed-line shows the approximate limit between sedimentary and crystalline bedrock. HGT: Hecla & Griper Trough. Light-gray lines: 100 m contours. Dashed black lines: Ice-stream lateral moraines on the sides of Sam Ford Trough. B. Location of the study area.**

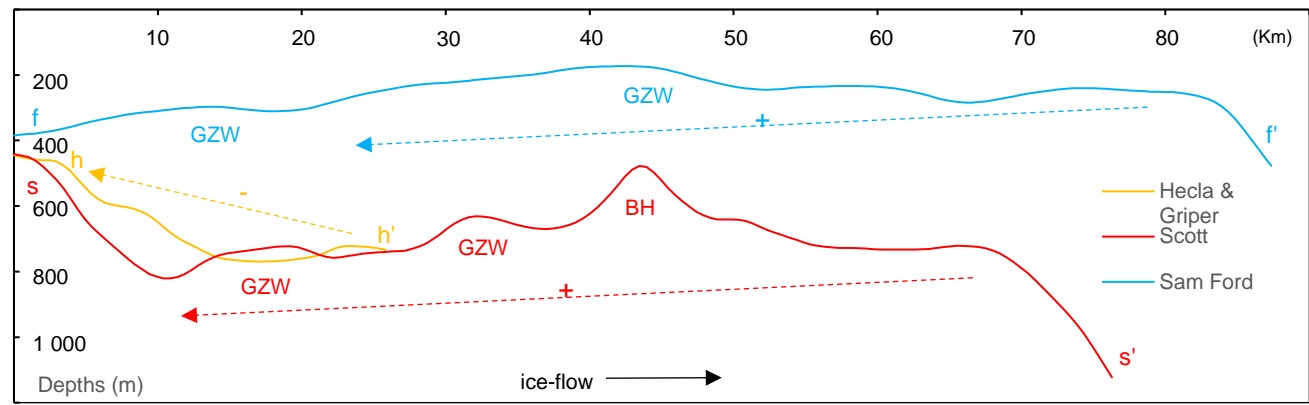

**Figure 2. A. Longitudinal depths profile along ice-flow route for each trough. Arrow with minus (-) symbol indicates a general up-ice decreasing profile of depths. Arrows with plus (+) symbol indicate a general up-ice increasing profile of depths (glacial overdeepening landward). BH: Bedrock high; GZW: Grounding-zone wedge.**

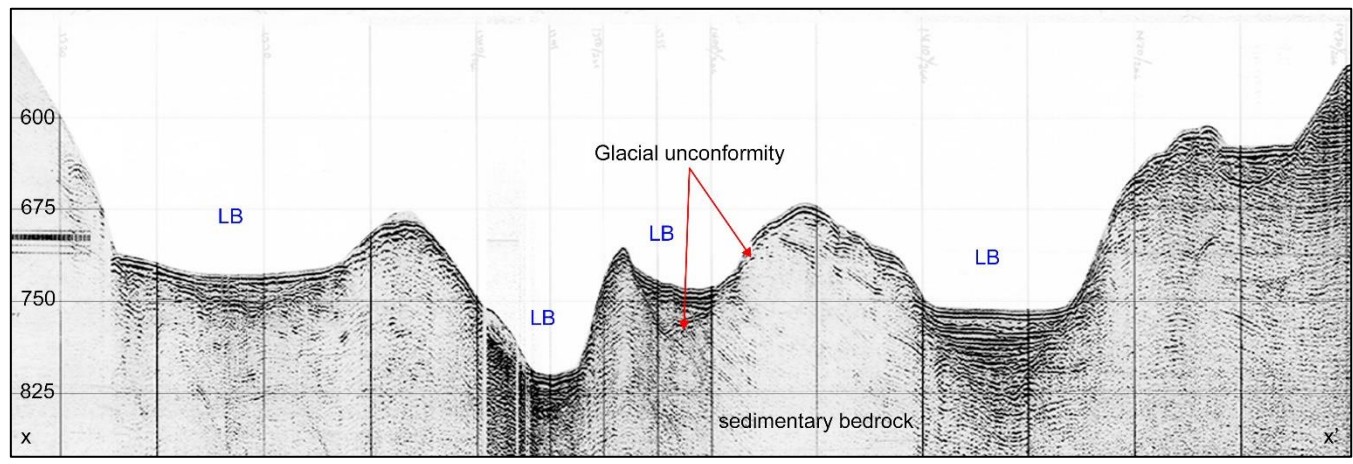

**Figrue 3. Airgun profile across Scott Trough showing glacially-eroded basins filled with sediments (LB) and residual bedrock ridges (Profile 78029_AG_266_0758; NRCan).**

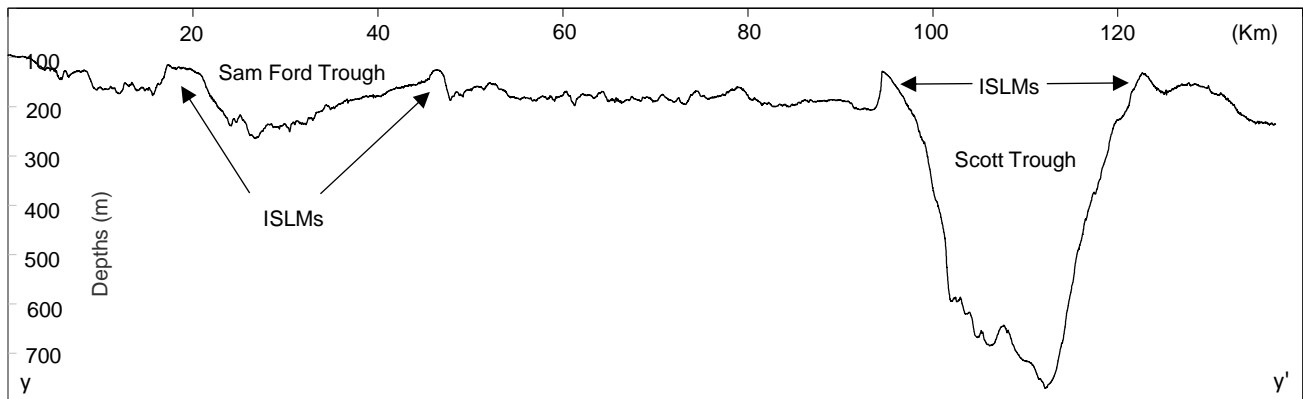

**Figure 4. Bathymetric profile across the northeastern Baffin Island Shelf showing Sam Ford and Scott troughs. The profile also shows the ice-stream lateral moraines on both sides of each trough.**

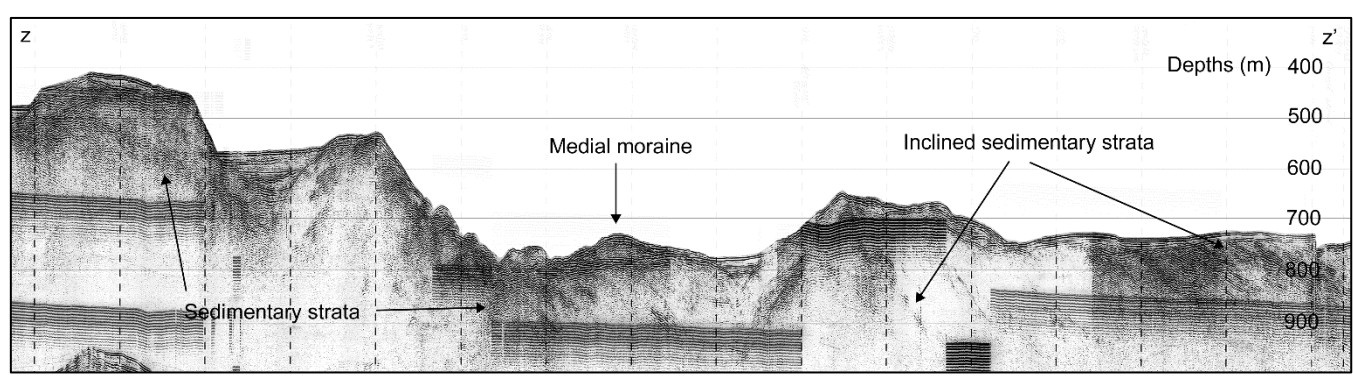

 **Figure 5. Airgun profile in Hecla & Griper Trough showing both sediment and bedrock ridges molded by glacial erosion (Profile 78029_AG_268_0110; NRCan). The profile also shows inclined sedimentary strata that were eroded by glacial ice.**

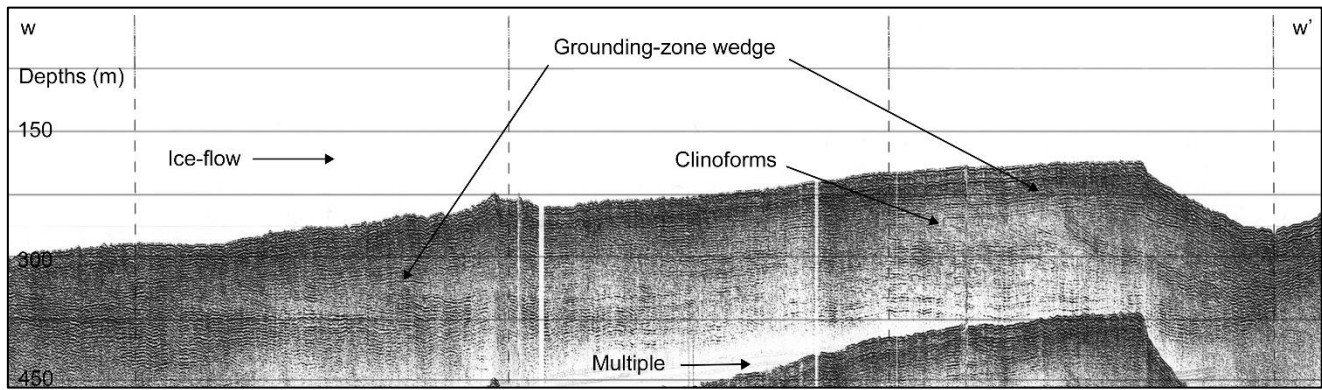

**Figure 6. Seismo-stratigraphic (Airgun) profile showing a major 75 m-thick grounding-zone wedge (red) in inner-middle Sam Ford Trough (Profile 80028_AG_RAYT_257_0200; NRCan).**

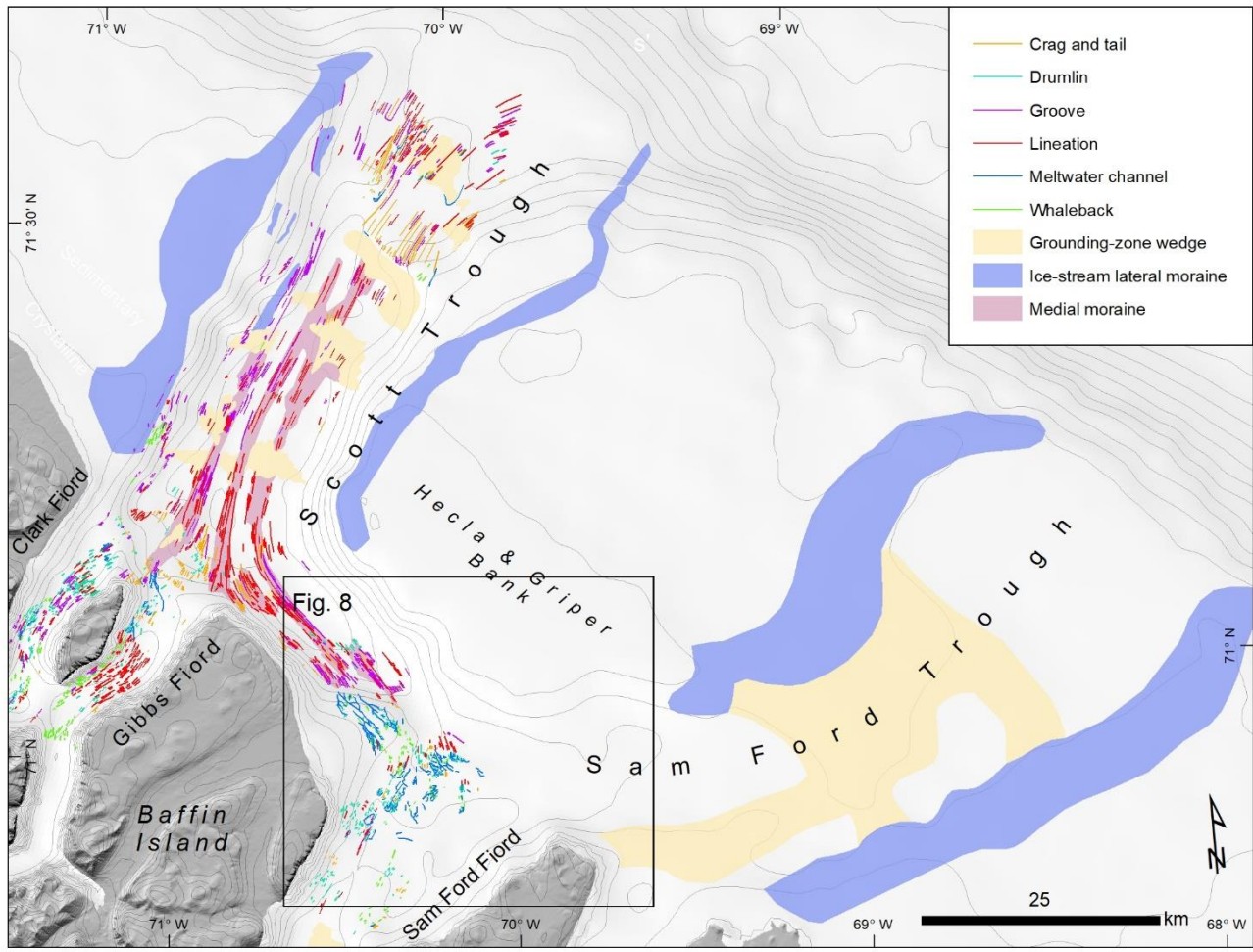

**Figure 7. Distribution of landforms related to ice-flow and to ice margin stabilization on the continental shelf and fiords of the study area, in northeastern Baffin Island, Eastern Arctic Canada.**

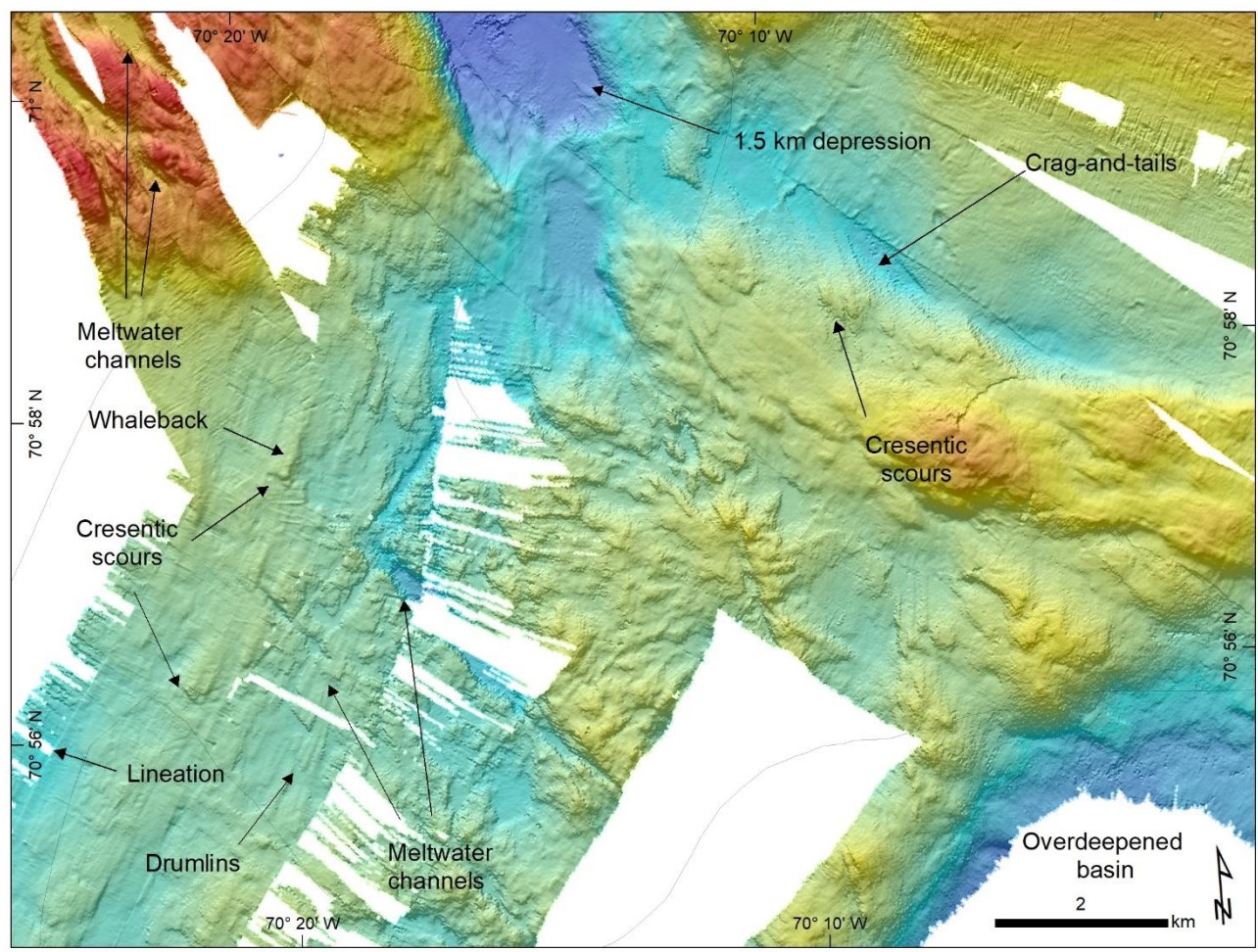

**Figure 8. Examples of ice-flow landforms on Sam Ford sill that are oriented towards Hecla & Griper Trough, Eastern Arctic Canada.**

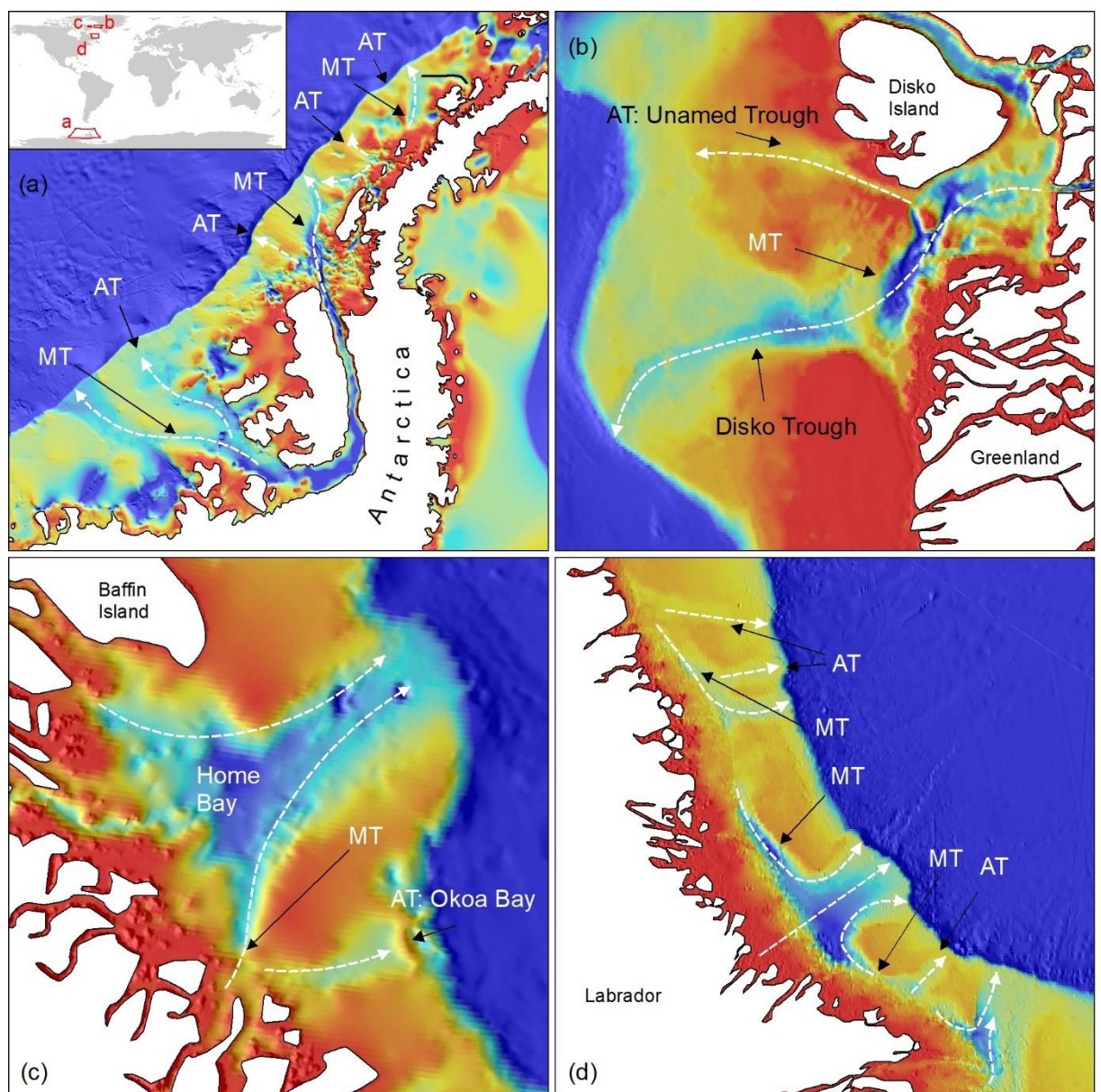

**Figure 9. A.** Bathymetry (BEDMAP2; Fretwell et al., 2013) of the western continental shelf of the Antarctic Peninsula. MT: Marginal trough. AT: Abandoned trough. White dashed lines: Interpreted ice stream tracks. Inset: Location of figures A-D. **B.** Bathymetry (IBCAO) of Unnamed and Disko troughs off West Greenland. MT: Marginal trough. **C.** Bathymetry (IBCAO) of Okoa Bay, Home Bay and marginal troughs off eastern Baffin Island. **D.** Bathymetry (IBCAO) of Labrador Shelf troughs off Labrador, Canada.
