# Peer review of "Ice-stream flow switching by up-ice propagation of instabilities along glacial marginal troughs"

_The Cryosphere, 2018_

## Referee Comment (RC1) · Anonymous Referee #1 · 18 Oct 2018

The authors provide a nice manuscript that describes a case study for the evolution of ice stream flow along the northeastern Laurentide Ice Sheet margin. The inference of ice stream piracy is based on the authors' interpretation of shelf bathymetry – chiefly how a prominent marginal (coast parallel) trough extends to a neighboring fjord. This sets up a topographic pattern such that two major fjords (the primary fjord plus the neighboring fjord) feed into a single cross-shelf trough that lies outboard of the primary fjord. The cross-shelf trough fronting the neighboring fjord is much shallower than the cross-shelf trough of the primary fjord. Thus the formation of the shallower cross-shelf trough is interpreted to be relict from prior to the ice piracy.

[Figure]

For the most part the manuscript clearly explains the authors' interpretation, although in my opinion it should be couched more as such – an interpretation – and not quite as factual as the authors tend to make it seem. The paper is well illustrated. The writing is mostly good; I provide some comments that would help clean it up a little. The manuscript is also fairly brief. I do not think that this is any sort of serious flaw – the contribution comes across as a note, or a case study, and I think that is fine. However, I found the manuscript to be lacking in its description of the glacial history of the study region and in covering the published literature on the evolution of fjords, which seems relevant to this topic (even though these are offshore troughs). I would suggest that the authors consider adding some text to their manuscript on these two topics (glacial history and fjord evolution).

I have one additional comment of note regarding the authors' main interpretation of ice piracy and flow switching. It strikes me that the authors' present one interpretation, and perhaps a very reasonable one given the likely nature of troughs to propagate in the headward direction over repeated glaciations. That said, can one really rule out an alternative? One alternative being that the topographic situation has always been the same as it is today – that the Hecla and Griper marginal trough funneled ice into the Scott system from day 1, and that only during the most extensive glaciations did Sam Ford ice spill onto the shelf fronting the entrance to Sam Ford Fiord. This led to a less developed (shallower) cross shelf trough and lack of a trough mouth fan. I think unless one drilled the Sam Ford Trough and found ancient tills, but not recent tills, one couldn't rule out this alternative scenario. For this reason, I really encourage the authors to use looser terms, couching their piracy mechanism as an "interpretation" or "inference". . .

Line by line comments:

Title. I wonder if a title that more closely describes the inferences made in the paper (e.g., using words such as "evolution of shelf troughs," or "ice piracy and flow switching") would better advertise this nice case study.
Line 13. Similar to what?

Overall abstract. Use of "marginal trough." I did not know what this was until it was defined later in the paper. Reading the abstract again made more sense after I learned the terminology. Given the broad audience of this journal, I wonder if the term can be defined – or a more descriptive term could be used – in the abstract?

Line 16. Suggest "Ice-flow switching was first invoked. . ." instead of "Ice-flow switching has first been invoked. . ."

Line 19. Can do without the word "then"

Line 25/26. I do not think that because a variety of driving mechanisms for flow switching have been described in the literature, that this means that the problem is poorly understood. All of the mechanisms might be valid in their various contexts. I also suggest that the sentence starting with "This wide range . . . " should be removed.

Paragraph beginning on Line 35. Are the terms "Scott Trough" and "Sam Ford Trough" formally defined – do they appear on topographic/bathymetric charts? If not, and are defined here, authors should point out that they are "informally" called. . .

The spelling of the word "fjord." It is fine to use the Scandinavian spelling when referring to fjords generally, but when referring specifically to the particular fjords on Baffin, they are proper place spelled "Fiord."

Line 38. It would be more correct to write that the MSGLs and a till unit that extend onto the TMF "have been interpreted" to indicate that the LIS reached the shelf break during the LGM (B and L, 2017). The evidence points to LGM ice cover, but it has not been confirmed with coring and dated sediments, for example?

Line 40. Sam Ford Fiord (not Fjord)

Line 42. Replace "Last glacial episode" with "LGM"

Line 42. Is it really possible that a glacier lobe on the continental shelf could have been

cold based? Would the bed have been below sea level or above? If below, I think cold-based is out of the question?

Line 42. Should write more literally. "...slow-flowing or cold-based ice" (not "slow flow... ice")

Line 43. "ice-flow" does not need a hyphen in this case.

Line 44. Look up use of "that" versus "which." Here and several instances throughout the manuscript the two are improperly used.

Line 45. This is a long and awkward – not entirely grammatically correct – sentence. I think there is a word or some words missing immediately in front of "bathymetric data are here used..." I found the info in the first half of the sentence to be too highly detailed about the methods to be appropriate for this paragraph.

Rather than "... used to analyze an ice stream switching..." I would suggest that what the authors are really doing is "hypothesizing" ice stream flow switching.

Line 54. "realized" awkward verb to use.

Line 58. "... was used to analyze Sam Ford Trough..." "data were"

Line 74. "...up to the middle of..." I can't make sense of this, what does "up to" mean, starting from the shelf break or the continent?

Line 75. Fiord (search and replace throughout manuscript, I won't comment again on these misspellings)

Line 77. 12 km long (not large)

Line 90-97. Starting with the sentence "If the tributary..." I become lost about the points being made. I think these sentences could be re-written. I also think that studies of fjord evolution, by headward erosion, should be discussed in this section.

Line 103. Here an anomalous high number of refs are used supporting the point.

Line 140. "flow" switching

Line 141. "Evidence" feels too strong for me. It is an interpretation. There are alternatives.

Line 159. The description of the "feedback" reminds me of the Kessler et al. (2008) paper on Baffin fiord evolution, which seems to be a quite relevant reference that doesn't appear in this manuscript.

Line 163. "an" ice sheet

Line 166. "in" both

Line 170. The discussion here seems to imply that ice piracy can take place within a single glacial cycle. I think it occurs over longer timescales? So I might argue that "surge" and "readvance" are not relevant here.

Line 170. I think a new paragraph could be started at "Although..."
* * *

---

## Referee Comment (RC2) · Anonymous Referee #2 · 6 Nov 2018

The manuscript of Brouard and Lajeunesse introduces an interesting topic that certainly merits discussion. However, as already noted by the editor, the manuscript is written in a short format style and has not been adapted to the regular format that is usually used in TC. Submissions to short format / high impact journals is a game we all sometimes play but, just for one's own self-respect, it shouldn't be obvious to the readers which manuscripts were originally submitted to journals other than where they ended up published. 'Dumping' a manuscript to a more field-specific journal without any changes – 'here you have it, if XY don't want it' – is disrespectful to readers. In addition, it misses a chance to make the manuscript a better paper, in this case to explain properly what has been done and to discuss the work against all the relevant

literature. I would therefore advise to redraft the text into a more rigorous manuscript format that would have the standard sections of methods, results, and discussion. The methods should explain what was done and why; at present they are just a brief list of the software used for the various steps of data manipulation.

In addition to a couple of minor points that I list further below, I have two comments on the interpretation of the geomorphology and the inferences made about the ice piracy both at the researched location and elsewhere. The authors state that 'the erosion and the morphology of the troughs of northeastern Baffin Shelf is a function of a competition for ice drainage basins' – while I agree with that statement, I consider their explanation, which infers a propagation of an instability wave upstream from the ice margin, as too simplistic. This is because the ice sheet is portrayed as a static feature through time while in reality its configuration was changing through time between very different states. Some ice configurations, such as an elongated ice field / ice cap / mountain ice sheet along the axis of Baffin Island favoured a denser ice drainage network draining small catchments via small ice streams, while the growth of the Foxe Dome and drainage of its ice across the high relief coast of Baffin Island likely favoured smaller number of larger ice streams. The analogue with the fluvial system is thus too simplistic. But I agree that once the Hecla & Griper Trough got eroded to its present depth, it prevented any ice being drained in the direction of the SF Trough. Ice streaming in the study area has been a subject of multiple studies (Briner et al., 2006, GSA B; De Angelis and Kleman, 2007, QSR; Briner et al., 2008, Geomorphology; Kessler et al., 2008, NG) and I find it unfortunate that none of these get any mention.

The authors state that 'ice piracy through the switching of ice streaming most probably occurred early during Pliocene-Pleistocene glaciations' in the case of the studied pair of cross-shelf troughs. By analogue, the authors infer that at other locations where similar trough configuration is observed, it might also be stemming from ice piracy. However, it's been shown that some parts of the continental shelf adjacent to high-relief coasts are composed largely of Quaternary sediments and have undergone a

much more complex dynamics than ice piracy through a few tributary ice streams (e.g. Montelli et al., 2017, MPG). The authors should thus indicate more clearly how deep is the transition between the crystalline bedrock and the sedimentary strata on the shelf and what is the thickness of the unconsolidated Quaternary sediments to rule out a more complex evolution of the drainage network than portrayed in their study. This in turn applies also to their generalisation about other similar trough configurations where a caveat should be inserted that unless it's been proven that the layer of Quaternary deposits is shallow, the inferences based on geomorphology might be incomplete and the longer term evolution of the ice drainage network might be recorded in the sedimentary sequences of the continental shelf.

Minor points: Line 18-19 two levels of parentheses not needed. Line 79 grounding zone wedges have been mentioned in the text before but the abbreviation has not been introduced. Line 110 Scott Ice Stream – check the spelling Line 136 No need to capitalise 'World' Agree with Reviewer 1 about the spelling of 'fiord'

Figures:

TC figure content guidelines state: 'Labels of panels must be included with brackets around letters being lower case (e.g. (a), (b), etc.).' 'Coordinates need a degree sign and a space when naming the direction (e.g. 30° N, 25° E).' This has not been followed.

I would advise to place a black frame around the map figures, which would also subdivide the individual figure panels. The frame should be the same thickness as the coordinate ticks, ideally 0.5 pt.

Fig. 2 The grounding zone wedge is supposed to be marked in red in panel A but I cannot see that. The location of the seismic profile, drawn in Fig. 1, should be noted in the Fig. 2 caption.

Move the letter A in panel A of Figure 4 somewhere out of the inset location panel. I suggest drawing the schemes of piracy on the shown DEMs in Figure 4. That would

help the reader to see the examples you note in the text. Okoa Bay not Okao Bay in Fig. 4C

---

## Editor Comment (EC1) · Stokes (Editor) · 30 Nov 2018

I would like to thank both reviewers for taking the time to provide two well-informed reviews. It is clear that they are generally supportive of publication, subject to some important revisions. I share their concerns about the brevity of the manuscript and would ask the authors to consider adding greater detail in terms of both the context and the results and discussion (if they decide to undertake revisions). As noted by one of the reviewers, this would also be facilitated by a more orthodox structure to the manuscript.

---

## Author Comment (AC1) · 23 Jan 2019

Etienne Brouard, Université Laval

Québec City, 23 Januray 2019

Dear Prof. Stokes, Please find enclosed (suppl.) our revised manuscript entitled " Ice-stream switching by up-ice propagation of instabilities along glacial marginal troughs" co-authored by Etienne Brouard and Patrick Lajeunesse, submitted for publication in the journal The Cryosphere.

All the comments and concerns are addressed in the responses-to-comments letter.

We greatly appreciate these comments that allowed us to improve this manuscript.

The data and conclusions presented in this paper have never been published. All authors have approved the submission of the manuscript. This manuscript contains 7654 words and 9 figures.

Best regards,

RESPONSE TO THE DECISION LETTER

First, we would like to thank the editor and the reviewers for insightful and valuable comments. Our revisions are described as follows.

EDITOR: Comments to the Author: I would like to thank both reviewers for taking the time to provide two well-informed reviews. It is clear that they are generally supportive of publication, subject to some important revisions. I share their concerns about the brevity of the manuscript and would ask the authors to consider adding greater detail in terms of both the context and the results and discussion (if they decide to undertake revisions). As noted by one of the reviewers, this would also be facilitated by a more orthodox structure to the manuscript.

Response: We thank the editor for the positive comment. We did revise the manuscript to improve the text and figures while incorporating most of the suggestions proposed by the reviewers. We did change the manuscript to a more standard structure (introduction, regional setting, methods, results, discussion, conclusions) and added details in all the sections to better support the interpretations. Below lies the point-by-point response to the reviewers.

REVIEWER 1:

Comment 1: For the most part the manuscript clearly explains the authors' interpretation, although in my opinion it should be couched more as such – an interpretation – and not quite as factual as the authors tend to make it seem. The paper is well illustrated. The writing is mostly good; I provide some comments that would help clean it

up a little.

Response 1: We acknowledge that some parts of the text are based on interpretations and we tried to make it clearer in the text where inferences and interpretations were made. We have changed the title to better explain that we found a probable cause (interpretation) for the morphology of the trough system.

Comment 2: The manuscript is also fairly brief. I do not think that this is any sort of serious flaw – the contribution comes across as a note, or a case study, and I think that is fine

Response 2: Following comments by the editor and reviewer 2, we did change the manuscript to a more standard structure (introduction, regional setting, methods, results, discussion, conclusions) and added various details in all the sections to better support the interpretations.

Comment 3: However, I found the manuscript to be lacking in its description of the glacial history of the study region and in covering the published literature on the evolution of fjords, which seems relevant to this topic (even though these are offshore troughs). I would suggest that the authors consider adding some text to their manuscript on these two topics (glacial history and fjord evolution).

Response 3: We have added a glacial history into a regional setting section (lines 61-85). We have also added some text on fiord/trough/overdeepening evolution, mainly on headward erosion processes, in the section Erosion of Hecla & Griper Trough to support the interpretations (lines 266-290).

Comment 4: I have one additional comment of note regarding the authors' main interpretation of ice piracy and flow switching. It strikes me that the authors' present one interpretation, and perhaps a very reasonable one given the likely nature of troughs to propagate in the headward direction over repeated glaciations. That said, can one really rule out an alternative? One alternative being that the topographic situation has

always been the same as it is today – that the Hecla and Griper marginal trough funneled ice into the Scott system from day 1, and that only during the most extensive glaciations did Sam Ford ice spill onto the shelf fronting the entrance to Sam Ford Fiord. This led to a less developed (shallower) cross shelf trough and lack of a trough mouth fan. I think unless one drilled the Sam Ford Trough and found ancient tills, but not recent tills, one couldn't rule out this alternative scenario. For this reason, I really encourage the authors to use looser terms, couching their piracy mechanism as an "interpretation" or "inference". . .

Response 4: We acknowledge concerns raised by the reviewer about alternative scenarios that would fit the morphology of the studied trough system. We have added alternative scenarios, including the one suggested by the reviewer in section 4.3 and explain why they cannot account for the trough morphologies (lines 297-315). We could, however, suggest a scenario where ice was partitioned between the two troughs, but that the up-ice propagation of ice streaming is still the main driver (lines 316-318).

Comment 5: Title. I wonder if a title that more closely describes the inferences made in the paper (e.g., using words such as "evolution of shelf troughs," or "ice piracy and flow switching") would better advertise this nice case study.

Response 5: We changed the title to 'Ice-stream flow switching by up-ice propagation of instabilities along glacial marginal troughs' following the reviewer suggestions.

Comment 6: Line 13. Similar to what?

Response 6: Changed to: Trough systems that are formed of cross-shelf troughs intersected by marginal troughs. (line 21).

Comment 7: Overall abstract. Use of "marginal trough." I did not know what this was until it was defined later in the paper. Reading the abstract again made more sense after I learned the terminology. Given the broad audience of this journal, I wonder if the term can be defined – or a more descriptive term could be used – in the abstract?

Response 7: We added a brief description of marginal troughs just after the mention in the abstract: . . . through glacial erosion of a marginal trough, i.e., deep parallel-to-coast bedrock moats located up-ice of cross-shelf troughs. (lines 13-14).

Comment 8: Line 16. Suggest "Ice-flow switching was first invoked. . ." instead of "Ice-flow switching has first been invoked. . ."

Response 8: Changed to Ice-flow switching was first invoked. (line 24)

Comment 9: Line 19. Can do without the word "then"

Response 9: Removed.

Comment 10: Line 25/26. I do not think that because a variety of driving mechanisms for flow switching have been described in the literature, that this means that the problem is poorly understood. All of the mechanisms might be valid in their various contexts. I also suggest that the sentence starting with "This wide range . . . " should be removed.

Response 10: We have changed the sentence to: The wide range of possible driving mechanisms outlines the fact that ice stream switching is a complex process that requires further assessment in order to model accurately the future behavior of modern ice sheets. (lines 33-34)

Comment 11: Paragraph beginning on Line 35. Are the terms "Scott Trough" and "Sam Ford Trough" formally defined – do they appear on topographic/bathymetric charts? If not, and are defined here, authors should point out that they are "informally" called. . .

Response 11: Scott Trough is defined on nautical charts of the Canada Hydrographic Survey as the extension of Scott Inlet. Scott Inlet is the toponym for the opening of Clark and Gibbs fjords to the Baffin Bay and therefore does not represent the bathymetric depression we analyse. Sam Ford Trough is defined as a prolongation of Hecla & Griper Trough on Nautical charts. However, Scott and Sam Ford Trough are informal names that are widely used in literature to represent the bathymetric depressions. These informal names were used for describing the bathymetric depressions for over

40 years and are still used today. We do not think adding this precision would benefit the text (e.g., Gilbert, 1985, Arctic; Osterman & Neilson 1989, CJES; Praeg et al. 2007, CGC, Bennett et al. 2014, Bull. Can. Petr. Geol., Brouard & Lajeunesse 2017, Scientific Report, Margold et al. 2015, Journal of Maps, 2018, QSR; Jenner et al. 2018, Marine Geology).

Comment 12: The spelling of the word "fjord." It is fine to use the Scandinavian spelling when referring to fjords generally, but when referring specifically to the particular fjords on Baffin, they are proper place spelled "Fiord."

Response 12: Every occurrence of 'fjord' has been changed to 'fiord'.

Comment 13: Line 38. It would be more correct to write that the MSGLs and a till unit that extend onto the TMF "have been interpreted" to indicate that the LIS reached the shelf break during the LGM (B and L, 2017). The evidence points to LGM ice cover, but it has not been confirmed with coring and dated sediments, for example?

Response 13: Jenner et al. (2018; Marine Geology) confirmed with cores along the slope off the shelf edge that the Laurentide Ice Sheet did reach the shelf edge as suggested (Brouard & Lajeunesse, 2017). We have added the Jenner et al. (2018) reference to support the point. (line 240)

Comment 14: Line 42. Replace "Last glacial episode" with "LGM"

Response 14: Changed to : The last glacial stage (MIS2) reached its maximum around 25 cal. ka BP in Western Baffin Bay (line 72)

Comment 15: Line 42. Is it really possible that a glacier lobe on the continental shelf could have been cold based? Would the bed have been below sea level or above? If below, I think cold-based is out of the question?

Response 15: That is a really good comment, as small parts of the continental shelf (Hecla & Griper Bank) may have been over sea-level during full glacial conditions, at least during the last glaciation. The inter fiord areas were characterized by cold-based

areas (Briner et al. 2006, GSA Bull.) and it is reasonable to believe that part of the ice in between the troughs could have as well be in some parts cold-based. However, we acknowledge that cold-based ice is unlikely. We removed cold-based ice.

Comment 16: Line 42. Should write more literally. ". . .slow-flowing or cold-based ice" (not "slow flow. . . ice")

Response 16: Changed to : slow-flowing ice (line 76).

Comment 17: Line 43. "ice-flow" does not need a hyphen in this case

Response 17: Hyphen removed.

Comment 18: Line 44. Look up use of "that" versus "which." Here and several instances throughout the manuscript the two are improperly used.

Response 18: We revised every occurrence of 'which' and 'that' in the text so that they are properly used.

Comment 19: Line 45. This is a long and awkward – not entirely grammatically correct – sentence. I think there is a word or some words missing immediately in front of "bathymetric data are here used. . ." I found the info in the first half of the sentence to be too highly detailed about the methods to be appropriate for this paragraph. Rather than "... used to analyze an ice stream switching. . ." I would suggest that what the authors are really doing is "hypothesizing" ice stream flow switching.

Response 19: This sentence was split in two and we added that we interpret an ice stream switch from the data.: Here, high-resolution swath bathymetry imagery combined with archived seismic reflection data and International Bathymetric Chart of the Arctic Ocean bathymetric data (IBCAO; Jakobsson et al., 2012) are used to analyze the morphology and stratigraphy of a single glacial trough network on the northeastern Baffin Shelf, in Eastern Arctic Canada. These data suggest that past ice stream switching occurred due to the ice-discharge piracy from an ice stream in a deep cross-shelf trough (Scott Trough) via the lateral extension of a marginal trough (Hecla and

Griper Trough), which led to the shutdown of the ice stream occupying the neighboring cross-shelf trough (Sam Ford Trough). (lines 44-48)

Comment 20: Line 54. "realized" awkward verb to use

Response 20: The whole sentence was changed to: Individual landforms were digitalized on the surface in ArcMap 10.2. . .. (line 96)

Comment 21: Line 58. ". . . was used to analyze Sam Ford Trough. . ." "data were"

Response 21: Changed was to were.

Comment 22: Line 74. ". . .up to the middle of. . ." I can't make sense of this, what does "up to" mean, starting from the shelf break or the continent?

Response 22: Change to: . . .grounding wedges in the trough. . .

Comment 23: Line 77. 12 km long (not large)

Response 23: Changed large to "wide".

Comment 24: Line 90-97. Starting with the sentence "If the tributary. . ." I become lost about the points being made. I think these sentences could be re-written. I also think that studies of fjord evolution, by headward erosion, should be discussed in this section.

Response 24: This is now in section 4.2, lines 266-290. Sentences were rewritten in order to better explain that the morphology of Hecla & Griper Trough is a function of Scott Trough morphology rather than the morphology of Sam Ford Fiord. We explain it by headward erosion. We also discuss headward erosion throughout the section.

Comment 25: Line 103. Here an anomalous high number of refs are used supporting the point

Response 25: Less relevant references were removed.

Comment 26: Line 140. "flow" switching

Response 26: Added "flow".

Comment 27: Line 141. "Evidence" feels too strong for me. It is an interpretation. There are alternatives.

Response 27: Changed to: ... provide for the first time an empirical context where adjacent ice streams on a continental shelf could have interplayed in a competition for ice discharge ... (lines 365-366)

Comment 28: Line 159. The description of the "feedback" reminds me of the Kessler et al. (2008) paper on Baffin fiord evolution, which seems to be a quite relevant reference that doesn't appear in this manuscript

Response 28: Kessler et al. (2008) paper was added throughout the text to support key concepts.

Comment 29: Line 166. "in" both

Response 29: Added "in".

Comment 30: Line 170. The discussion here seems to imply that ice piracy can take place within a single glacial cycle. I think it occurs over longer timescales? So I might argue that "surge" and "readvance" are not relevant here.

Response 30: The erosion of the through is a long-term process, while the switch is a point in time where ice discharge change direction. So, during the glacial cycle when the switch occurred, the winning ice stream gains mass balance and thus can equilibrate by advancing its margins. Changed to : The merging of ice streams through ice piracy should result in an increase in ice discharge and erosion rates in the "winning" trough. The winning ice stream gains mass balance and thus should equilibrate by advancing its margins (if it is not already at the shelf break). (lines 393-396)

Comment 31: Line 170. I think a new paragraph could be started at "Although. . ."

Response 31: A new paragraph was started at line 397.

REVIEWER 2

Comment 1: The manuscript of Brouard and Lajeunesse introduces an interesting topic that certainly merits discussion. However, as already noted by the editor, the manuscript is written in a short format style and has not been adapted to the regular format that is usually used in TC. Submissions to short format / high impact journals is a game we all sometimes play but, just for one's own self-respect, it shouldn't be obvious to the readers which manuscripts were originally submitted to journals other than where they ended up published. 'Dumping' a manuscript to a more field-specific journal without any changes – 'here you have it, if XY don't want it' – is disrespectful to readers. In addition, it misses a chance to make the manuscript a better paper, in this case to explain properly what has been done and to discuss the work against all the relevant literature. I would therefore advise to redraft the text into a more rigorous manuscript format that would have the standard sections of methods, results, and discussion. The methods should explain what was done and why; at present they are just a brief list of the software used for the various steps of data manipulation.

Response 1: We apologize for the minimum changes that were made on the original manuscript. We did change the manuscript to a more standard structure (introduction, regional setting, methods, results, discussion, conclusions) and added details in all the sections to better support the interpretations. We added what was done and why in the methods section.

Comment 2: In addition to a couple of minor points that I list further below, I have two comments on the interpretation of the geomorphology and the inferences made about the ice piracy both at the researched location and elsewhere. The authors state that 'the erosion and the morphology of the troughs of northeastern Baffin Shelf is a function of a competition for ice drainage basins' – while I agree with that statement, I consider their explanation, which infers a propagation of an instability wave upstream from the ice margin, as too simplistic. This is because the ice sheet is portrayed as a static feature through time while in reality its configuration was changing through time

between very different states. Some ice configurations, such as an elongated ice field / ice cap / mountain ice sheet along the axis of Baffin Island favoured a denser ice drainage network draining small catchments via small ice streams, while the growth of the Foxe Dome and drainage of its ice across the high relief coast of Baffin Island likely favoured smaller number of larger ice streams. The analogue with the fluvial system is thus too simplistic. But I agree that once the Hecla & Griper Trough got eroded to its present depth, it prevented any ice being drained in the direction of the SF Trough.

Response 2: We added a discussion on fiord/trough formation showing other studies that found headward (up-ice) erosion to be a main driver in the erosion of overdeepened basins (section 4.2). The headward erosion creates feedbacks favoring ice-channeling and ice-flow acceleration. Channelization in troughs in flat areas such as continental shelves is likely to result into a smaller number of ice streams alike on Baffin Island Shelf. The headward erosion is very similar to channelization in fluvial systems. We acknowledge that the analogue of the fluvial system is probably too simplistic to explain every occurrence of marginal troughs on formerly glaciated shelves. However, we cannot rule out that the headward erosion is most likely to result into a system similar to fluvial captures. For those reasons we maintain the fluvial analogue but added . . . An ice-drainage piracy mechanism similar to river captures in fluvial systems can probably not explain the occurrence of all the other abandoned cross-shelf troughs on most high-latitude continental shelves that do not extend up-ice to the coast or to fiords, mainly because each have different geological, glaciological and climatic contexts. However, the presence of morphologically similar systems can be observed on many formerly glaciated shelves: e.g,. . .(lines 347-350).

Comment 3: Ice streaming in the study area has been a subject of multiple studies (Briner et al., 2006, GSA B; De Angelis and Kleman, 2007, QSR; Briner et al., 2008, Geomorphology; Kessler et al., 2008, NG) and I find it unfortunate that none of these get any mention.

Response 3: We incorporated the reference suggested by the reviewers in the text to

better support some interpretations.

Comment 4: The authors state that 'ice piracy through the switching of ice streaming most probably occurred early during Pliocene-Pleistocene glaciations' in the case of the studied pair of cross-shelf troughs. By analogue, the authors infer that at other locations where similar trough configuration is observed, it might also be stemming from ice piracy. However, it's been shown that some parts of the continental shelf adjacent to high relief coasts are composed largely of Quaternary sediments and have undergone a much more complex dynamics than ice piracy through a few tributary ice streams (e.g. Montelli et al., 2017, MPG). The authors should thus indicate more clearly how deep is the transition between the crystalline bedrock and the sedimentary strata on the shelf and what is the thickness of the unconsolidated Quaternary sediments to rule out a more complex evolution of the drainage network than portrayed in their study. This in turn applies also to their generalisation about other similar trough configurations where a caveat should be inserted that unless it's been proven that the layer of Quaternary deposits is shallow, the inferences based on geomorphology might be incomplete and the longer term evolution of the ice drainage network might be recorded in the sedimentary sequences of the continental shelf.

Response 4: We added the thickness (< 100 m) of unconsolidated Quaternary sediments on the Baffin Shelf in Section 2. We added a caveat at the of the beginning (see response to comment 2) and at the end of the generalization to other glaciated continental shelves: However, further investigations are needed to confirm that competition between ice stream played a role in the development of these trough systems. (lines 359-360).

Comment 5: Minor points: Line 18-19 two levels of parentheses not needed

Response 5: This was a citation software problem. We revised the manuscript in order to correct similar errors.

Comment 6: Line 79 grounding zone wedges have been mentioned in the text before

but the abbreviation has not been introduced.

Response 6: We added the abbreviation at the first mention.

Comment 7: Line 110 Scott Ice Stream – check the spelling

Response 7: We revised the manuscript for any spelling errors.

Comment 8: Line 136 No need to capitalise 'World'

Response 8: Changed to world.

Comment 9: TC figure content guidelines state: 'Labels of panels must be included with brackets around letters being lower case (e.g. (a), (b), etc.).' 'Coordinates need a degree sign and a space when naming the direction (e.g. 30ᵃŮę N, 25ᵃŮę E).' This has not been followed.

Response 9: We revised thoroughly all the figures so that they follow all of TC figure content guidelines.

Comment 10: I would advise to place a black frame around the map figures, which would also subdivide the individual figure panels. The frame should be the same thickness as the coordinate ticks, ideally 0.5 pt.

Response 10: Black frames and coordinate thicks are now all black and have all a thickness of 0.5 points.

Comment 11: Fig. 2 The grounding zone wedge is supposed to be marked in red in panel A but I cannot see that. The location of the seismic profile, drawn in Fig. 1, should be noted in the Fig. 2 caption.

Response 11: Red was removed.

Comment 12: Move the letter A in panel A of Figure 4 somewhere out of the inset location panel. I suggest drawing the schemes of piracy on the shown DEMs in Figure 4. That would help the reader to see the examples you note in the text. Okoa Bay not

[Figure]

Okao Bay in Fig. 4C

Response 12: The letter was moved outside of the inset location. We draw white arrows showing the assumed ice stream tracks. Okoa Bay has been corrected.

Please also note the supplement to this comment:
https://www.the-cryosphere-discuss.net/tc-2018-196/tc-2018-196-AC1-supplement.pdf
* * *
[Figure]

**Supplement:**

[revised manuscript text omitted]

---

## Editor Comment (EC2) · Stokes (Editor) · 30 Jan 2019

I would like to thank the authors for their careful consideration of the reviewer comments and for submitting a revised manuscript. Given the substantial nature of the revisions and the addition of new material, I have decided to send this back out to one reviewer.

---

## Author Response (AR2)

**Etienne Brouard**

Université Laval

Chris R. Stokes

5   Editor

The Cryosphere

Québec City, 27 February 2019

10   Dear Prof. Stokes,

Please find enclosed our revised manuscript entitled *" Ice-stream switching by up-ice propagation of ice-streaming along glacial marginal troughs"* co-authored by Etienne Brouard and Patrick Lajeunesse, submitted for publication in the journal *The Cryosphere.*

15   All the comments and concerns are addressed in the responses-to-comments letter. We greatly appreciate these comments that allowed us to improve this manuscript.

The data and conclusions presented in this paper have never been published. All authors have approved the submission of the manuscript. This manuscript contains 7479 words and 9 figures.

Best regards,

25   **Etienne Brouard, Ph.D.**

Département de géographie

Université Laval

Québec, QC Canada G1V 0A6

etienne.brouard.1@ulaval.ca

30   Tél.: (418)-473-4501

**RESPONSE TO THE DECISION LETTER**

First, we would like to thank the editor and the reviewers for insightful and valuable comments. Our revisions are described
35 as follows.

**EDITOR:**
*Comments to the Author:*
I have now received a further referee report on your revised manuscript and I'm pleased to inform you that they are generally
40 very satisfied. They have, however, raised some concerns about the large number of relatively minor errors/typos in the text.
I have also read through the revised manuscript and would encourage you to attend to their list of suggested edits and also
give the manuscript a final, thorough, proof-read.

*Response:*
45 We did revise thoroughly the manuscript to improve the text and to remove any typos/small errors. Below lies the point-by-
point response to the reviewers. Other corrections are detailed in the marked-up version of the manuscript.

50 **REVIEWER 1:**
*Comment 1:*
Following the advice from the first round of review, the authors have revised their manuscript into a regular format. I think
the paper benefits from that and I consider the study well laid out now, with clearly described objectives, methods, results,
and their implications.
55 I would recommend the authors to give the manuscript at least one more proper read—there is a lot of typos and omissions.
The authors might also consider having the text proof-read by a colleague to further work on the style—the phrasing is a
little weird at places and it does not read all that well.

*Response:*
60 Following comments by the editor and the reviewer, we did thoroughly revise the manuscript to remove any typos and
improve the style.

*Comment 2:*
L 18     " … competition for ice discharge between ice streams, which implies piracy of ice-drainage basins via marginal
65 troughs…" This is a weird statement: competition for ice discharge between ice streams does not have to automatically
imply piracy of drainage basins through via marginal troughs.

*Response:*
We did specify that the competition that implied ice piracy was between the two ice streams of the study area.
70 Changed to: These results suggest that competition for ice discharge between the two ice streams, which implies piracy of
ice-drainage basins via marginal troughs, was the driving mechanism behind ice flow-switching. (lines 17–19)

*Comment 3:*
L19-20 "the union of ice catchment by piracy" I suggest to possibly rephrase it to "the enlargement of its ice catchment by
75 piracy…"

*Response:*
Changed union to enlargement (line 19)

80 *Comment 4:*
L 25 West Antarctic Ice Sheet

*Response:*
Changed Antarctica to: Antarctic (line 25)

**Comment 5:**
L 34–35 "data …. are needed" (data is the plural of datum)

*Response:*
Changed is to: are (line 35)

**Comment 6:**
L 64–65 "The glacial overdeepening of the troughs probably began during the late Pliocene"—up until some ten years ago, the extent of glaciation on the shelf was debated for the LGM, one of the coldest periods of the Pleistocene. In the Late Pliocene the ocean was significantly warmer and the cold parts of the climate fluctuations were shorter and less pronounced. The cross shelf troughs might be younger than the fiords: the troughs being a product of fully-fledged ice sheets while the fiords might have been repeatedly incised by the gradually growing forms of glaciation in the Late Cenozoic.

*Response:*
We acknowledge that the fiords are probably older features than the troughs. Since ~1 Myr is needed to produce well-developed fiords (Kessler et al. 2008) we removed any references to the Pliocene.
Change to: The erosion of the troughs is intrinsically linked to the erosion of the deep fiords of northeastern Baffin Island, which modeling suggests were eroded to present-day depths in ~1 Myr (Kessler et al., 2008). Therefore, the glacial overdeepening of the troughs probably began during the Pleistocene and erased all traces of preglacial fluvial systems (Løken & Hodgson, 1971). (lines 62–65)

**Comment 7:**
L 68: Ice streams in plural

*Response:*
Corrected.

**Comment 8:**
L 67–69: The sentence is weirdly worded after "i.e.", I don't fully understand.

*Response:*
Changed to: To produce well-developed fiords, the position of these ice streams was probably stable throughout most of the Pleistocene, i.e., limited to fiords and troughs. (lines 66–68)

**Comment 9:**
L 75: Either extended to the shelf break or reached the shelf break, having both is unnecessarily complicated.

*Response:*
We have changed the sentence to: During the MIS2 (25 – 16 ka BP), Scott and Hecla & Griper troughs were inundated by ice streams of the LIS (Briner et al., 2006b; De Angelis & Kleman, 2007; Margold et al., 2015b; Brouard & Lajeunesse, 2017) that extended to the shelf break at the mouth of the troughs, while Sam Ford Trough was occupied slow-flowing ice (Brouard & Lajeunesse, 2017). (lines 72–74)

**Comment 10:**
L 77      There's some uncertainty on the dating so it would be better to write "until ca. 14.1 cal ka BP". Also, I find it unnecessary to write "cal. ka"—in general context, "ka" is now used for absolute/calendar time not for radiocarbon.

*Response:*
Following the reviewer suggestion, we removed mentions of "cal." in this paragraph.

135

*Comment 11:*
L 80      Not "up to" but "from as early as"

*Response:*
140    Changed to: as early as (line 78)

*Comment 12:*
L 81      the presence of

145    *Response:*
Added "the" before presence (line 79)

*Comment 13:*
L 90 "The specifics of acquisition" consider rewording

150
*Response:*
Changed to: The metadata (line 88)

*Comment 14:*
155    L 95      "The complete surface was transferred in ESRI ArcMap 10.2 software geomorphological mapping and topographic analyses." Something is missing in that sentence.

*Response:*
Added "for" before geomorphological mapping and topographic (line 94) "for geomorphological mapping and for
160    topographic analyses"

*Comment 15:*
L 96–98  : Change to "Individual landforms were digitised in ArcMap 10.2 and interpreted based on their apparent character (width, length, orientation, etc.) and relevant literature (e.g., Dowdeswell et al., 2016b)."

165
*Response:*
Changed to: Individual landforms were digitised in ArcMap 10.2 and interpreted based on their apparent character (width, length, orientation, etc.) and relevant literature (e.g., Dowdeswell et al., 2016b).' (lines 94–96)

170    *Comment 16:*
L 125      marked by a steep wall

*Response:*
Added "by" before "a steep wall" (line 122)
175
*Comment 17:*
L 130      delete one Fiord

*Response:*
180    Deleted. (line 127)

***Comment 18:***
L 144    Delete 's" in "Troughs"

*Response:*
Deleted.

***Comment 19:***
L 173 The term "wet bed glaciers" is somewhat uncommon (Google Scholar only gives five hits)—I would recommend to change it to "warm-based glaciers".

*Response:*
Changed to: warm-based glaciers (line 169)

***Comment 20:***
L 179 lineations

*Response:*
Removed the typo.

***Comment 21:***
L 184    "Cross-cut by grounding-zone wedges" this phrasing is somewhat unfortunate. It migh be better to write that the areas with MSGLs are at places overprinted/covered by GZW or something in that sense but it's not really a cross-cutting relationship.

*Response:*
Change to: Where mega-scale glacial lineations are covered by grounding-zone wedges, lineations were interpreted to reflect time-transgressive ice flows occurring during the landward retreat of an ice stream (lines 178–180)

***Comment 22:***
L 188    Like instead of alike

*Response:*
Changed to: As with … (line 183)

***Comment 23:***
L 193 King first than Jezek in the REF

*Response:*
This was due to a wrong Copernicus referencing style. Every reference was corrected to be in chronological order rather than in alphabetical order.

***Comment 24:***
L 243    in the same way

*Response:*
Changed 'Accordingly, the medial moraine ridges are interpreted to be the product of differential ice-stream erosion. In a same way, coalescence of multiple ice streams probably favored the formation of subglacial medial moraines over the ridges' to: Accordingly, the medial moraine ridges are interpreted to be the product of differential erosion between multiple ice streams coalescing. (lines 235–236)

*Comment 25:*
L 270 'steering along was'—something missing in that sentence?

235    *Response:*
Changed to: 'steering alone was' (line 262)

*Comment 26:*
L 332–333: Delete the repeated citation.

240
*Response:*
Deleted.

*Comment 27:*
245    L 391    ' … and leads"

*Response:*
Changed 'provoked' to:  leads (line 381)

250    *Comment 28:*
L 411 provide

*Response:*
Corrected.
255
*Comment 29:*
Formatting comment:
Order of references in the in-text citations: it should be ordered chronologically not alphabetically

260    *Response:*
This was due to a wrong Copernicus referencing style. The order of references was corrected to be in chronological order rather than in alphabetical order.

[revised manuscript text omitted]